# Oncometabolite D-2-hydroxyglutarate—dependent metabolic reprogramming induces skeletal muscle atrophy during cancer cachexia

Xinting Zhu[1,7], Juan Hao[2,7], Hong Zhang[1], Mengyi Chi[1], Yaxian Wang[1], Jinlu Huang[1], Rong Xu[1], Zhao Xincai[1], Bo Xin[1], Xipeng Sun[1], Jianping Zhang[1], Shumin Zhou[3], Dongdong Cheng[4], Ting Yuan [4], Jun Ding[5], Shuier Zheng[6], Cheng Guo [1✉] & Quanjun Yang [1✉]

Cancer cachexia is characterized by weight loss and skeletal muscle wasting. Based on the up-regulation of catabolism and down-regulation of anabolism, here we showed genetic mutation-mediated metabolic reprogramming in the progression of cancer cachexia by screening for metabolites and investigating their direct effect on muscle atrophy. Treatment with 93 µM D-2-hydroxyglutarate (D2HG) resulted in reduced myotube width and increased expression of E3 ubiquitin ligases. Isocitrate Dehydrogenase 1 (*IDH1*) mutant patients had higher D2HG than non-mutant patients. In the in vivo murine cancer cachexia model, mutant *IDH1* in CT26 cancer cells accelerated cachexia progression and worsened overall survival. Transcriptomics and metabolomics revealed a distinct D2HG-induced metabolic imbalance. Treatment with the *IDH1* inhibitor ivosidenib delayed the progression of cancer cachexia in murine GL261 glioma model and CT26 colorectal carcinoma models. These data demonstrate the contribution of *IDH1* mutation mediated D2HG accumulation to the progression of cancer cachexia and highlight the individualized treatment of *IDH1* mutation associated cancer cachexia.

[1] Department of Pharmacy, Shanghai Sixth People's Hospital affiliated Shanghai Jiao Tong University School of Medicine, 600 Yishan Road, Shanghai 200233, China. [2] Department of Endocrinology, Shanghai Traditional Chinese Medicine, Integrated Hospital, Shanghai University of Traditional Chinese Medicine, 230 Baoding Road, Shanghai 200082, China. [3] Institution of microsurgery on extremities, Shanghai Sixth People's Hospital affiliated Shanghai Jiao Tong University School of Medicine, 600 Yishan Road, Shanghai 200233, China. [4] Department of Bone Oncology, Shanghai Sixth People's Hospital affiliated Shanghai Jiao Tong University School of MedicineShanghai Shanghai, Shanghai, P. R. China. [5] Department of Neurosurgery, Shanghai Sixth People's Hospital affiliated Shanghai Jiao Tong University School of Medicine, 600 Yishan Road, Shanghai 200233, China. [6] Department of Oncology, Shanghai Sixth People's Hospital affiliated Shanghai Jiao Tong University School of Medicine, 600 Yishan Road, Shanghai 200233, China. [7]These authors contributed equally: Xinting Zhu, Juan Hao. ✉email: guopharm@126.com; myotime@sjtu.edu.cn

Cancer cachexia is a multifactorial wasting syndrome characterized by progressive weight loss and persistent erosion of host body cell mass in response to a malignant growth[1,2]. Cancer cachexia occurs in 50%-80% of cancer patients and is an independent predictor of poor prognosis[3]. Moreover, cachexia is associated with reduced treatment tolerance, therapeutic response, quality of life, and short survival. The ongoing loss of skeletal muscle mass contributes to progressive weight loss. Animal experiments suggest that preservation of skeletal muscle mass delays the process of cachexia and prolongs survival[4,5]. Currently, there are few treatments available to preserve skeletal muscle mass as a treatment for cachexia. Clinical evidence has shown that some types of solid tumors appear to be predisposed to cachexia[6].

The progressive loss of skeletal muscle mainly occurs due to increased catabolism and decreased anabolism[1]. Metabolic reprogramming is common in cancer patients[7]. Several metabolites and metabolic pathway changes have been observed in cancer patients with cachexia[8–11]. Metabolic byproducts accumulate due to the excessive growth of the host tumor and a dysfunctional metabolism. Studies have showed that genome instability and gene mutations also contribute metabolic reprogramming of cancer cachexia by regulating gene expression[12].

These altered metabolites not only accumulated due to the limited metabolic disposal capacity of cachexia, but also displayed extensive pathological and physiological functions. Oncometabolites were known to contribute to tumourigenesis, angiogenesis, progression, and metastasis[13]. However, little is known about whether genetic mutation-mediated metabolic reprogramming can influence the initiation, progression, and treatment of cancer cachexia. For example, the oncometabolite D2-hydroxyglutarate (D2HG) is produced by mutant isocitrate dehydrogenase (IDH)1/2. IDH1 mutations are found in several types of cancer. D2HG is specifically elevated in cancer patients, including glioma, chondrosarcoma, acute myeloid leukaemia, intrahepatic bile-duct cancer, and angioimmunoblastic T-cell lymphoma[12,14–19]. D2HG has been reported to be associated with myopathy[12,20]. However, the role of these cancer related metabolites in muscle metabolism remains unknown.

The present study investigated cancer-related metabolites in the development of cachexia by screening for active metabolites and examining their effect on muscle wasting. The effect of metabolite mediated muscle wasting was confirmed by oncogene mutation and high metabolite treatment. Furthermore, downstream metabolic enzyme gene was overexpressed in in vitro experiment to reveal the mechanism, and enzyme inhibitor was used in in vivo experiments for pharmacodynamic research of cancer cachexia individualized treatment.

## Results

### The oncometabolites D2HG and fumarate induce muscle atrophy.
After consulting published articles[8–11], we listed cancer cachexia-related metabolites and identified 157 cachexia-related metabolites (Supplementary Data 1). The metabolites were aligned with the Human Metabolome Database (www.hmdb.ca) and the functional annotations were included. To exclude metabolites that changed in only one project, we found and selected 66 common metabolites that showed consistent changes in multiple research projects (Fig. 1a). Based on the muscle-related functional annotation and the accessibility of these metabolites, nineteen candidate metabolites were selected for the in vitro experiment (Supplementary Data 2). The concentrations of these metabolites were listed based on the reference listed in Supplementary Data 2.

Well differentiated myotubes were treated with these metabolites for in vitro screening, and myotube diameters were measured

(Fig. 1b, c). Fumarate (Fum), lactate, D2HG, pyruvate, adenosine, inosine, carnosine, phenylacetate, 1-methylhistidine, 3-methylhistidine, 4-hydroxyproline, and creatine induced different degrees of myotube atrophy. To confirm the effect of metabolite mediated muscle wasting, we used the widely used in vitro muscle atrophy system and used mRNA expression of Trim63 (MuRF1) and Fbxo32 (Atrogin-1) upregulation as an index of muscle proteolysis[21]. When well-differentiated myotubes were treated with these 19 metabolites for 48 h, treatment with succinate, fumarate, D2HG, 1-methylhistidine, 3-methylhistidine, and 4-hydroxyproline resulted in a significant increase in mRNA expression of Trim63 and Fbox32 (Fig. 1d, e).

Genome instability and gene mutations in cancer cells drive the gain or loss of certain enzyme functions. On the other hand, they exert pro-oncogenic capabilities through metabolic reprogramming[22]. We searched for the oncogenes and common cancer related genetic alterations in the database of CancerGenetics Web (http://www.cancerindex.org/geneweb/), Cancer Gene Census(https://cancer.sanger.ac.uk/census), oncogene database (http://ongene.bioinfo-minzhao.org/), and database of Mutational Signatures (https://cancer.sanger.ac.uk/cosmic). A total of 2739 genes were included in the database (Supplementary Data 3). Genotype/phenotype/disease associations were entered based on a comprehensive biological context for OMICs data interpretation[23]. To screen for the metabolism related genes, 84 genes were selected based on the association between gene annotation and metabolism (Supplementary Data 4). We then searched for the association between these genes and metabolic phenotype. It was found the SDH and IDH1 were associated with the metabolites fumarate and D2HG. Therefore, we further investigated the IDH1-mediated D2HG accumulation and SDH-mediated fumarate (Fum) accumulation (Fig. 1f). We treated the C2C12 myotube with D2HG and Fum for 72 h and confirmed the morphological changes. Total RNA was then extracted for transcriptome sequencing. Heatmaps showed distinct transcriptional characteristics after D2HG treatment based on fragments per kilobase of exon model per million mapped fragments (FPKM) (Fig. 1g). The Fum treatment and NTC groups showed similar transcriptional characteristics. These results confirmed that D2HG-induced myotube wasting occurs by proteolysis and with distinct transcriptional characteristics.

### IDH1 mutation mediates high levels of D2HG in cancer patients with cachexia and in an in vivo cachexia mouse model.
D2HG is produced by mutant IDH1[15], a unique R132H/C/G mutation at rs121913500 (Supplementary Fig. S1a). We obtained the IDH1 mutation information from patients, and nineteen out of 149 cancer patients were confirmed to have an IDH1 mutation by PCR sequencing (Supplementary Data 5). Serum D2HG levels were then measured using high-performance liquid chromatography-tandem mass spectrometry. Serum D2HG levels were higher in IDH1 mutant patients compared to controls (Fig. 2a). This was consistent with previous studies showing abnormally elevated levels of D2HG to millimolar per gram of tissue in patients with a single mutant copy of IDH1[6]. We defined cancer cachexia as weight loss >5% in the last 6 months or weight loss >2% in the last 6 months and a body mass index <20 kg/m$^2$. Of 19 IDH1-mutated patients, 8 had cancer cachexia. Serum levels of D2HG were higher in patients with cancer cachexia compared to patients with stable-weight cancer (Fig. 2b).

Next, we evaluated the IDH1 genotype-related cancer phenotypes based on the public biological information resources of the TCGA database based on cBioPortal[24]. The average IDH1 alteration frequency of IDH1 was 5% across all cancer, and the most common alteration type was mutation (Supplementary

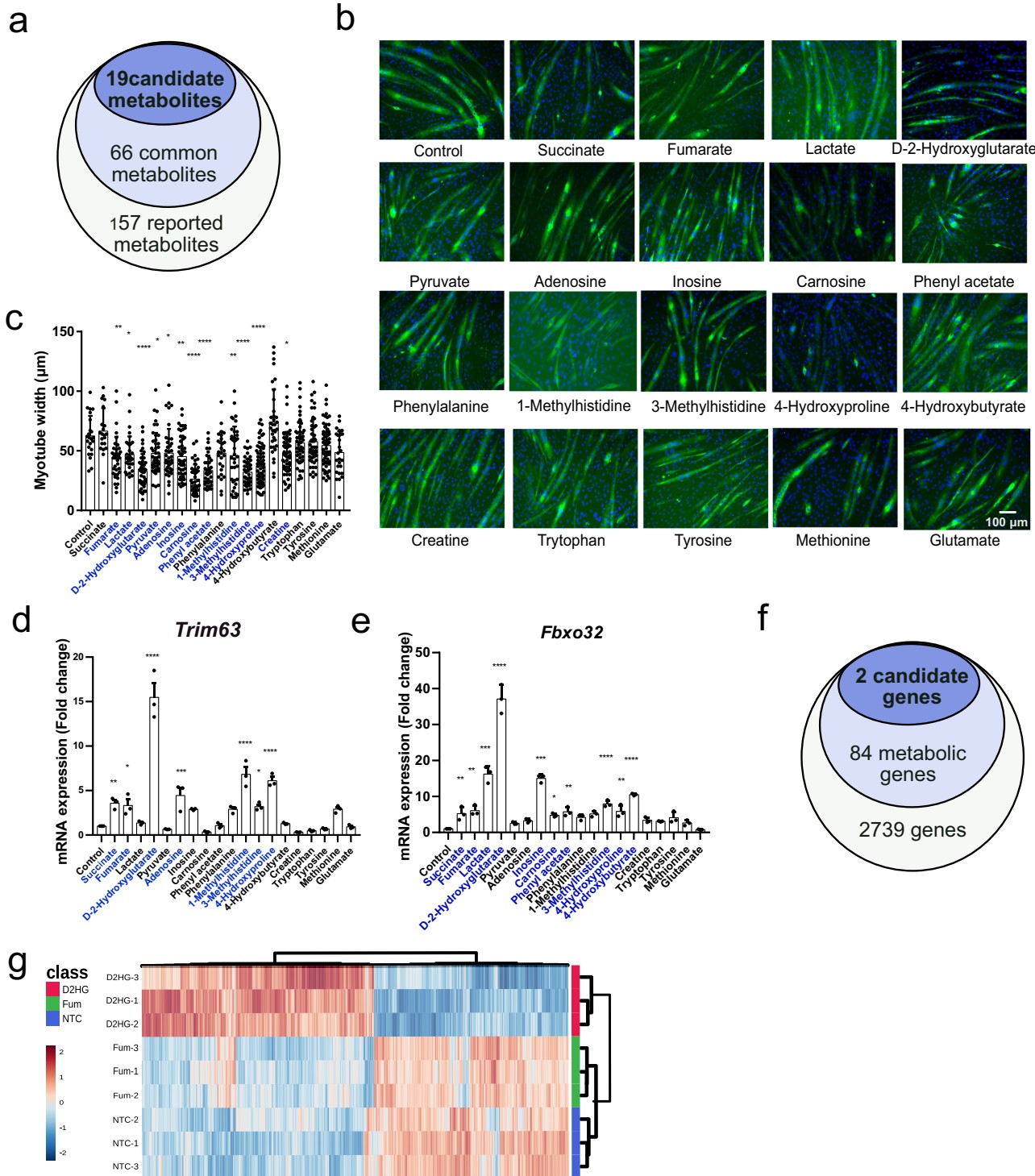

**Fig. 1 Screening of cachexia-related metabolites and identification of metabolites that induced muscle wasting. a** Flowchart of the multistep selection of candidate metabolites related to muscle wasting. **b** Typical immunofluorescence morphological changes after metabolite treatments. Blue indicates DAPI, and green indicates MyHC (scale bar: 100 μm). **c** Differences in myotube diameters between different metabolite treatments. Each dot represents 1 myotube diameter. One-way ANOVA with post hoc Tukey's multiple comparison tests (****$p < 1e-4$, ***$p < 2e-4$, **$p < 2e-3$, *$p < 0.05$). **d**, **e** Identification of metabolites that induce high gene expression of the E3 ligase *Trim63* and *Fbxo32*. One-way ANOVA with post hoc Tukey's multiple comparison tests (****$p < 1e-4$, ***$p < 2e-4$, **$p < 2e-3$, *$p < 0.05$). **f** Screening of common genes associated with altered cancer based on a Venn diagram. **g** Heatmap of differentially expressed genes after D2HG and Fum treatment of well-differentiated myotubes compared to NTC myotubes. Each group was tested in triplicate. Gene expression values are expressed as fragments per kilobase per million.

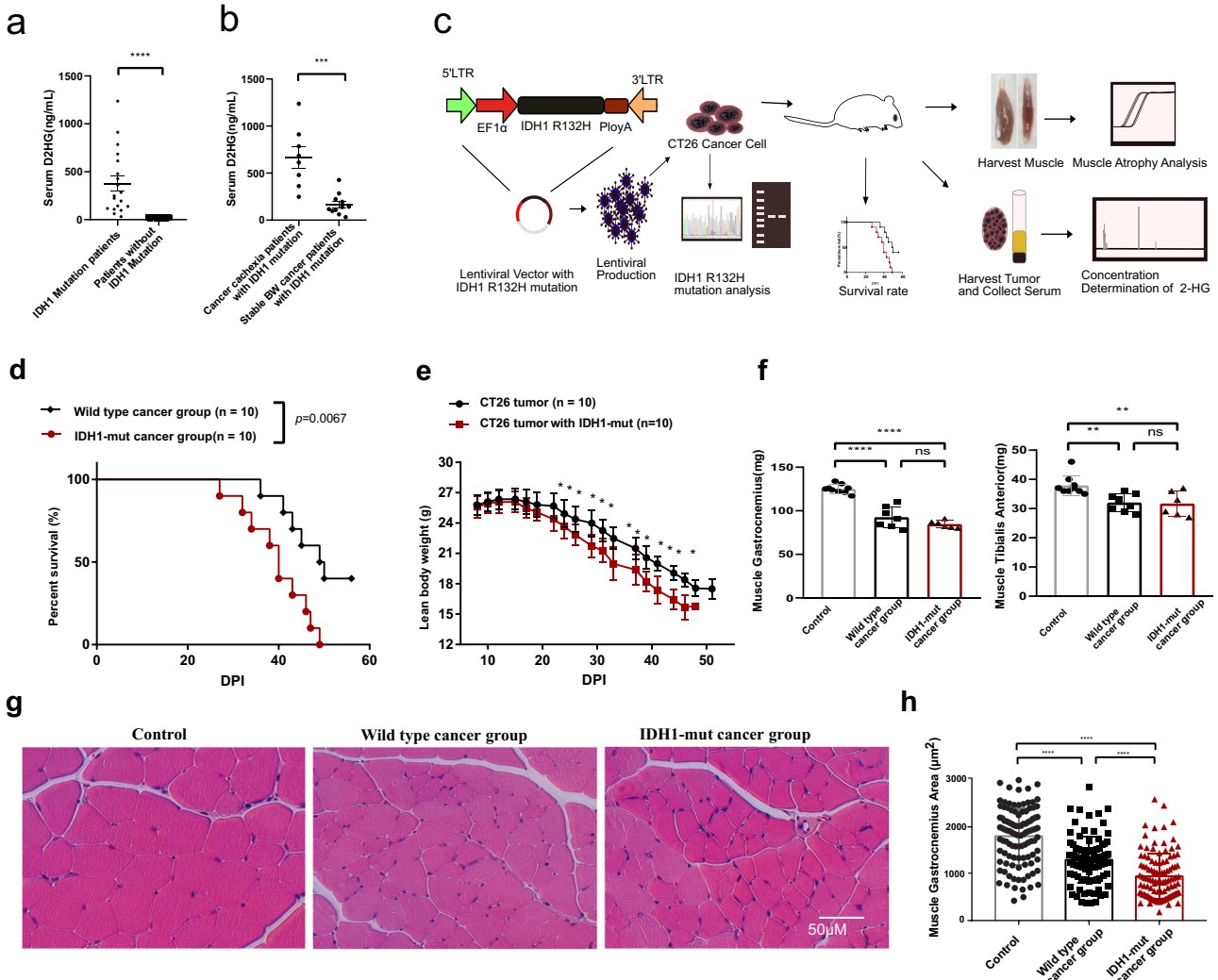

**Fig. 2 *IDH1* mutation-mediated D2HG accumulation in cancer patients and in vivo experiment of D2HG mediated muscle atrophy. a** Column plot of serum D2GH levels in 194 cancer patients with/without *IDH1* mutation. Unpaired two-tailed *t*-test (****$p < 1e-4$, ***$p < 2e-4$, **$p < 2e-3$, *$p < 0.05$). **b** Column plot of serum D2GH levels in cancer cachexia patients with an *IDH1* mutation and in cancer patients with stable body weight with an *IDH1* mutation. Unpaired two-tailed *t*-test (****$p < 1e-4$, ***$p < 2e-4$, **$p < 2e-3$, *$p < 0.05$). **c** Schematic of the in vivo experimental design for *IDH1* (R132H) mutation in CT26 tumor-induced skeletal muscle atrophy and cancer cachexia syndrome. **d** Survival curve of mice with *IDH1* (R132H) mutant cancer and wild-type cancer. Log-rank (Mantel-Cox) test (****$p < 1e-4$, ***$p < 2e-4$, **$p < 2e-3$, *$p < 0.05$). **e** Lean body weight (whole body weight - tumor weight) of the *IDH1* (R132H) mutant cancer and wild-type cancer. Unpaired two-tailed *t*-test. (*$p < 0.05$ *IDH1* (R132H) mutant cancer vs. wild-type cancer). **f** Column plot of the muscle gastrocnemius and anterior tibial muscle in *IDH1* (R132H) mutant cancer-bearing mice and wild-type cancer-bearing mice. One-way ANOVA with post hoc Tukey's multiple comparison test (****$p < 1e-4$, ***$p < 2e-4$, **$p < 2e-3$, *$p < 0.05$). **g** Typical cross-section histopathological image of the muscle gastrocnemius (scale bar: 50 μm). **h** Column plot of the cross-section area of muscle gastrocnemius in *IDH1* mutant cancer bearing mice and wild-type cancer-bearing mice. One-way ANOVA with post hoc Tukey's multiple comparison test (****$p < 1e-4$, ***$p < 2e-4$, **$p < 2e-3$, *$p < 0.05$).

Data 6). The substitution of the arginine 132 by histidine (R132H) accounted for >80% of all IDH mutations. To evaluate the overall survival of *IDH1* alteration in pan-cancer, a total of 10,802 patients from 32 studies with mutation data were included after excluding 10 overlapping patients. A total of 627 patients with *IDH1* alterations showed a worse overall s *IDH1* alteration frequency of *IDH1* urvival compared to patients without *IDH1* mutation (Supplementary Fig. S1b). Although IDH mutations have a different prognostic value depending on the cancer type, we still found that the *IDH1* mutation mediated poor survival in pan-cancer patients. Furthermore, the transcripts per million in different cancers ranged from 20 to 200, and were higher in cancer patients than in controls (Supplementary Fig. S1c). We therefore tested the hypothesis that *IDH1* mutation-mediated

D2DH accumulation contribute to the progression of cancer cachexia.

The *IDH1* mutation at the amino acid arginine 132 (R132) is unique because it is localized in the substrate binding site of the isozyme[14]. *IDH1* catalyses the oxidative decarboxylation of isocitrate to ketoglutarate with the concomitant production of NADPH. When the *IDH1* mutation occurs in R132H, it hinders the hydrophilic interactions between the arginine and the α-carboxylates, and thus the mutant *IDH1* has gained a function that converts ketoglutarate and NADPH into D2HG and $NADP^+$. Based on the mutation frequency, we imported R132H into a cancer cell and used an in vivo experiment to evaluate whether *IDH1* mutation mediates D2HG accumulation during cancer cachexia progression (Fig. 2c). BALB/c mice

bearing CT26.wt colon adenocarcinoma cells are the most commonly used model of cancer cachexia[25]. We first cloned IDH1-R132H into the lentiviral plasmid pLV-EF1α-FLAG-IRES-Puro and produced pLV-EF1α-IDH1-R132H-FLAG-IRES-Puro lentiviruses, which were used to infect CT26 colon adenocarcinoma cells. IDH1 R132H amplification and protein expression were confirmed by cDNA sequencing and Western blot (Supplementary Fig. S2a, b). No IDH1 protein changes were observed in the wild-type group, whereas IDH1-R132H protein expression was higher in the IDH1-mutant group (Supplementary Fig. S2b). Consistent with the protein expression, the mRNA expression of IDH1-R132H was found to be higher (Supplementary Fig. S2c). Next, 2 million CT26 cancer cells were subcutaneously transplanted subcutaneously into the right flanks of male mice. The survival time of the mice with IDH1-mutant cancer was shorter than in control mice (Fig. 2d). There was a similar tumor weight curve, while the lean body weight of mice bearing CT26 tumor with IDH1 mutation was significant lower than mice bearing wild-type CT26 tumor from DPI 22. Cachexia was defined as a loss of more than 5% of the lean body weight (mice without transplanted tumor) from the lean body weight change curve. In IDH1-R132H mutant cancer-bearing mice, cancer cachexia syndrome occurred at DPI 17, and the average lean body weight decreased by 5.4% (from $26.83 \pm 1.12$ g to $25.45 \pm 0.98$ g) (Fig. 2e). However, cachexia was observed in the wild-type tumor group at DPI 22 as lean body weight decreased by 5.0% (from $27.09 \pm 1.10$ g to $25.81 \pm 1.23$ g). These results indicated that the mutation of IDH1 in CT26 cells could accelerate the tumor growth and induce body weight loss.

To determine the contribution of lean body weight loss, we measured typical skeletal muscle mass. Compared to the control group, the loss of skeletal muscle gastrocnemius and tibialis anterior in mice bearing IDH1-mutant cancer was 26.1% and 16.3%, respectively (Fig. 2f). Furthermore, the loss of muscle gastrocnemius was 10.6% between the IDH1-mutant cancer group and IDH-wt group. Since skeletal muscle accounts for ~40% of total body weight, skeletal muscle weight loss was the major contributor to body weight loss. From the histopathology results of the muscle gastrocnemius (Fig. 2g), IDH1-mutation in the tumor resulted in a smaller cross-sectional area in the muscle gastrocnemius compared to the control groups and non-mutated CT26 tumor-bearing cancer cachexia mice (Fig. 2h).

We then extracted total RNA from the muscle gastrocnemius and measured the expression of the E3 ligases. Trim63 and Fbxo32 expression was increased by 2.3 fold and 1.4 fold, respectively, in CT26-bearing cachexia mice without IDH1 mutation (Fig. 3a). In contrast, they were dramatically increased by 3.4 fold and 2.3 fold, respectively in muscle gastrocnemius of the CT26-bearing cancer cachexia mice with IDH1 mutation compared with controls, respectively. These results suggested that the degradation of skeletal muscle protein was enhanced through the ubiquitinated proteasome pathway (UPP). The protein of Atrogin-1 and MuRF1 were also increased in cancer cachexia group (Fig. 3b, c, and Supplementary Fig. S3a). Furthermore, the mice bearing IDH1-mutant tumor expressed higher Atrogin-1 and MuRF1 compared with mice bearing IDH1-wt tumor. The protein degradation of ubiquitin was determined by western blot also showed consistent results (Fig. 3d, and Supplementary Fig. S3b). The protein synthesis was reflected by the phosphorylation level of mTOR, p70s6k, and 4E-BP1 (Fig. 3e, f, and Supplementary Fig. S3c). Cachexia group showed low phosphorylation level of mTOR, p70s6k, and 4E-BP1, especially the mice bearing IDH1-mutant tumor. To reveal the mediator of IDH1R132H mutation in cancer, we measured the levels of total D2HG in serum and tumor tissues. D2HG was enriched in IDH1 mutant tumor and serum (Fig. 3g), which was consistent with

clinical data showing that IDH1 mutation at R132 resulted in a high concentration of D2HG and high frequency of cachexia in cancer patients. The mutant IDH1 in CT26 and GL261 cells mediated D2HG accumulation was evident from in vitro experiments (Supplementary Fig. S2d). These results confirmed that the deterioration of muscle atrophy was mediated by the high concentration of D2HG in CT26 bearing mice with an IDH1 mutation.

**D2HG induces proteolysis via up-regulation of the ubiquitinated protein system and metabolic reprogramming.** An ex vivo analysis of D2HG on differentiated multi-nucleus myotubes was designed (Fig. 4a). We treated well-differentiated multi-nucleus myotubes with 93 μM D2HG for 5 days, after which immunofluorescence was performed to detect myosin heavy chain (Supplementary Fig. S4a). The average myotube diameter of the D2HG treatment group was smaller than that of the NTC group (Supplementary Fig. S4b). To confirm the effect of protein degradation, we treated well-differentiated multinucleated myotubes with D2HG for 5 days and extracted total RNA. The mRNA expressions of E3 ligases of Trim63 and Fbxo32 were increased by ~11.7 and 20.4 fold, respectively (Supplementary Fig. S4c). In addition, the E2 ubiquitin-conjugating enzyme Ube2d1 was also upregulated by 2.8 fold (Supplementary Fig. S4d). These results indicate that the upregulated expression of UPP contributes to D2HG-induced muscle atrophy.

D2HG can regulate transcriptional and metabolic processes, such as glycolysis[22], lipogenesis, oxidative stress, and histone methylation[12]. We analyzed the transcriptional changes of D2HG-induced muscle atrophy and found distinct transcriptional profile (Fig. 4b). D2HG treatment resulted in 412 transcriptional changes based on fold change at 2.0 and adjusted q at 0.05 (Supplementary Data 7). To reveal the primary mechanism responsible for the catabolic pathway, overrepresentation analysis (ORA) was used to screen the different genes. ORA-based gene ontology (GO) enrichment revealed altered molecular function, biological process, and cellular component based on up- or down-regulated genes (Fig. 4c). Extracellular matrix structural constituent, structural molecule activity, structural constituent of the cytoskeleton, oxidoreductase activity, acting on the CH-OH group of donors NAD or NADP as acceptor, glutathione transferase activity, and NAD binding were the top molecular functions. We confirmed the concentration of NADH and NAD$^+$, it was interesting that D2HG-induced increased NAD$^+$/NADH redox ratio (Supplementary Fig. S4e). In addition, the biological process and cellular components, including muscle cell differentiation, muscle tissue development, muscle system process, and muscle contraction, were also disturbed after D2HG treatment (Supplementary Fig. S5a–d). We confirmed the correlation between muscle structure and metabolism resulting from D2HG treatment in differentiated multi-nucleus myotubes based on the GO enrichment analysis.

To reveal the distinct metabolic process responsible for the D2HG treatment, we extracted and measured the selected metabolites from well-differentiated myotubes using a targeted metabolomics strategy. Seventy-one metabolites were included based on the metabolic pathway of KEGG and previously reported metabolites [8–11]. The heat map showed a distinct separation of the D2HG treatment group (Fig. 4d). Joint Pathway Analysis was then used to integrate the altered metabolites and genes (Fig. 4e). The most characteristic pathways included glutathione metabolism, hypertrophic cardiomyopathy, aminoacyl-tRNA biosynthesis, citrate cycle synthesis, and degradation of ketone bodies. The altered metabolites were constructed based on the KEGG metabolic pathway (Fig. 4f), and we found

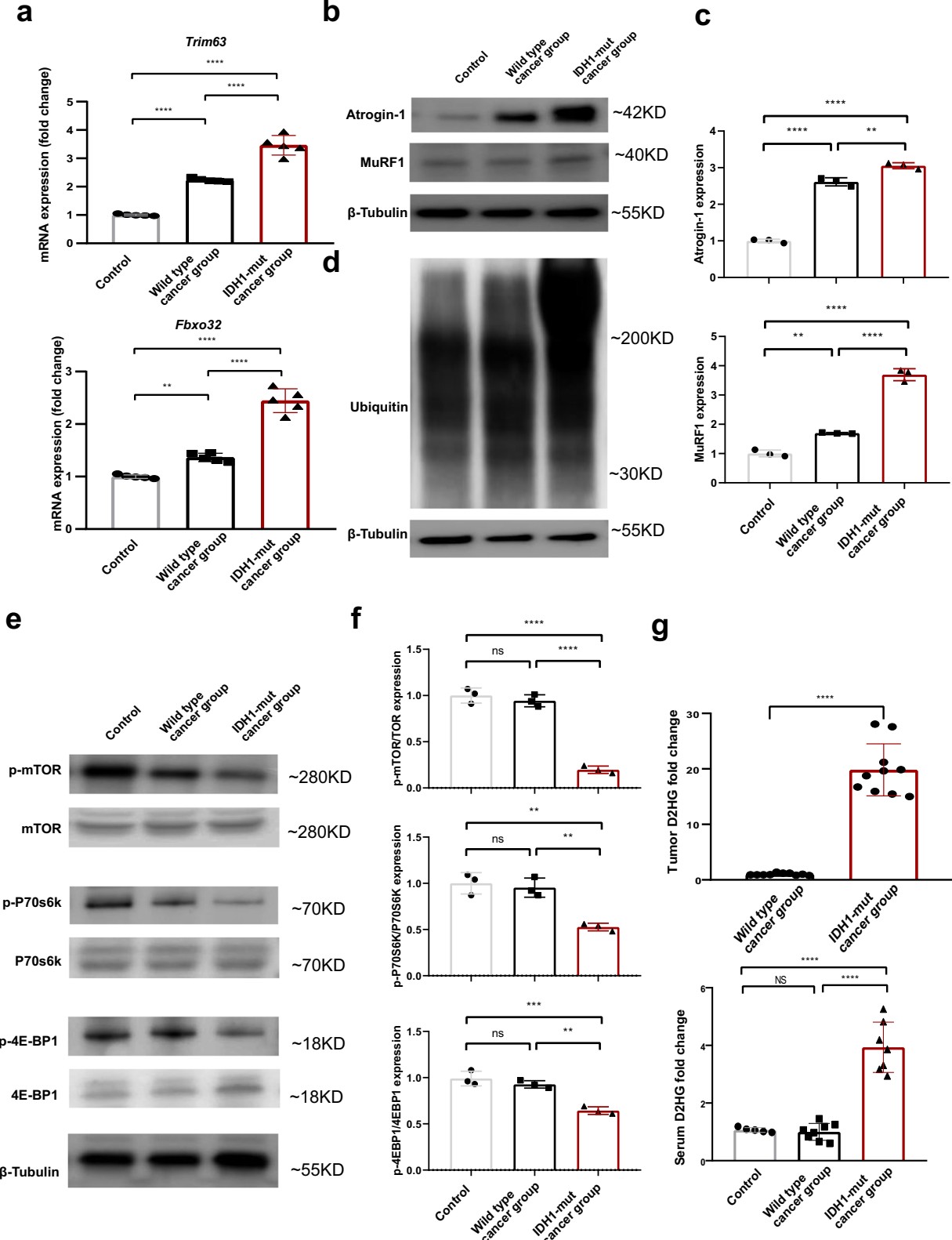

that D2HG exerted metabolic pathway reprogramming to disrupt the maintenance of metabolic homeostasis.

**D2hgdh mediated catabolism reverse D2HG-induced proteolysis**. We then analyzed the catabolic pathway of D2HG and investigated the effect of D2HG catabolism on the reversal of proteolysis and muscle atrophy. D2HG is catabolized by D2hgdh,

a mitochondrial enzyme encoding D-2hydroxyglutarate dehydrogenase. Mutation of this gene in humans has been associated with developmental delay, epilepsy, hypotonia, and dysmorphic features[26]. Since well-differentiated myotubes cannot catabolize D2HG, we cloned D2hgdh into C2C12 myoblasts and induced their differentiation into myotubes. Typical immunofluorescence of myosin heavy chain staining showed normal differentiation for

**Fig. 3 *IDH1* mutation-mediated D2HG accumulation induced a decrease in muscle protein synthesis and an increase in muscle protein proteolysis.**
**a** Column plot of Fbxo32 and Trim63 mRNA expression in *IDH1* mutant cancer-bearing mice and wild-type cancer-bearing mice. One-way ANOVA with post hoc Tukey's multiple comparison test (****$p < 1e-4$, ***$p < 2e-4$, **$p < 2e-3$, *$p < 0.05$). **b**, **c** The protein expression and quantification data of the degradation-related E3 ligases atrogin-1 and MuRF1 in *IDH1*-mutant cancer-bearing mice and wild-type cancer-bearing mice. One-way ANOVA with post hoc Tukey's multiple comparison test (****$p < 1e-4$, ***$p < 2e-4$, **$p < 2e-3$, *$p < 0.05$). **d** The protein expression of degradation-related ubiquitin in *IDH1* mutant cancer-bearing mice and wild-type cancer-bearing mice. **e**, **f** The protein expression and quantification data of synthesis-related mTOR, P70S6K and 4E-BP1 expression and phosphorylation levels in *IDH1* mutant cancer-bearing mice and wild-type cancer-bearing mice. One-way ANOVA with post hoc Tukey's multiple comparison test (****$p < 1e-4$, ***$p < 2e-4$, **$p < 2e-3$, *$p < 0.05$). **g** Column plot of tumor and serum D2HG levels in the *IDH1* mutant cancer bearing group and wild type cancer bearing mice. One-way ANOVA with post hoc Tukey's multiple comparison test (****$p < 1e-4$, ***$p < 2e-4$, **$p < 2e-3$, *$p < 0.05$).

D2hgdh overexpressing C2C12 myoblasts induced with a differentiation medium. However, there were sloppy myotubes after D2HG treatment (Fig. 5a). The myotube diameter of the D2HG-treated group was shorter than normal myotubes, whereas D2hgdh overexpression reversed the D2HG-induced myotube atrophy (Fig. 5b). We then measured the relative levels of D2HG and keto-glutarate in the differentiated myotubes. D2HG treatment resulted in high levels of D2HG and low levels of keto-glutarate (Fig. 5c). No differences were found in keto-glutarate, while the levels of D2HG were decreased, suggesting that the overexpression of D2hgdh in myotubes catabolizes D2HG. We also measured the mRNA expression of *Ube2d1, Trim63*, and *Fbxo32* (Fig. 5d, e). Still, D2HG-induced high expression of *Ube2d1, Trim63*, and *Fbxo32*, while D2hgdh overexpressing myotube inhibited the up-regulation of mRNA expression of these genes after D2HG treatment.

To confirm the expression of D2hgdh and related metabolic enzymes, we measured the protein expression by Western blot (Fig. 5f, and Supplementary Fig. S6). D2hgdh overexpression showed high expression of *D2hgdh, Hmgcr, Hsd17b7, Dhrs3*, and *Adh7*, but low expression of *IDH1*. Hmgcr is a rate-limiting enzyme for cholesterol synthesis that is regulated by a sterol-mediated negative feedback mechanism. Anti-HMGCR antibody-positive patients often have autoimmune myopathy and resemble limb-girdle muscular dystrophy[27]. Hsd17b7 regulates fatty acid metabolism and testosterone synthesis. Testosterone has a pronounced effect on muscle protein synthesis and muscle mass increase, especially during rapid muscle cell growth. A higher expression level of Hsd17b7 has been in broilers, which may result in a higher testicular mass required for early embryonic patterning[28]. Dhrs3 protein is required for early embryonic patterning and upregulation of Dhrs3 has been associated with osteogenic differentiation[29]. A single nucleotide polymorphism in *Adh7* was associated with multiple system atrophy[30]. Interestingly, myotubes overexpressing D2hgdh reversed the effect of D2HG. These results indicated that D2hgdh overexpression could enhance the catabolism of metabolite D2HG and subtract its proteolysis effect.

**D2hgdh reprogramming metabolism of D2HG involving in multiple processes**. HPLC-MS-MS-based targeted metabolomics was used to cluster the samples and individual metabolites (Fig. 5g). All four groups were categorised based on 60 metabolites, and 2HG levels were low in the NTC and D2hgdh groups. Myotubes overexpressing D2hgdh showed a similar metabolic profile to the NTC group. The D2HG treatment groups showed different metabolic characteristics, suggesting that D2HG may indicate a distinct metabolic change. Furthermore, D2HG treated myotubes overexpressing D2hgdh were clustered between the two groups, thus indicating that D2hgdh overexpression could resist D2HG-induced proteolysis. To directly show the metabolic pathway, we plotted the metabolic pathway and the relative concentration of metabolites (Supplementary Fig. S7). Of the

altered metabolites from two independent experiments, 12 were enriched, and 15 were simultaneously depleted (Fig. 5h and Supplementary Data 8).

To further reveal the transcriptional characteristics of the four groups, a heatmap was used to reveal the clustering profile as metabolomics (Fig. 5i). D2HG-induced distinct transcriptional features as the heatmap was clustered far away from the NTC group. Overexpression of D2hgdh could counteract D2HG-induced gene changes. The overall differentially expressed gene profile of the four groups showed that D2HG upregulated 1340 genes and downregulated 786 genes, while myotubes overexpressing D2hgdh showed 20 upregulated genes and 65 downregulated genes with a fold change of 2 and padj of 0.05 (Fig. 5j). This indicated implied that D2HG-induced a wide range of gene transcriptional level changes, while myotube overexpressing D2hgdh showed subtle perturbation. This set of experiments also included D2HG-treated well-differentiated myotubes and control myotubes (Supplementary Data 9). In addition, based on the repeatability assay, the Venn diagram revealed the distinct transcriptional alteration, showing that 37 genes were commonly up-regulated, and 101 genes were down-regulated after D2HG treatment (Fig. 5k and Supplementary Data 9).

To reveal the distinct transcriptional and metabolic properties of D2hgdh and to assess the reversal of D2HG-mediated metabolic reprogramming by D2ghdh overexpression, we used the pairwise comparison on D2hgdh overexpression myotubes. First, we compared D2ghdh overexpression in well-differentiated myotubes with control well-differentiated myotubes (Supplementary Data 10). The volcano plot showed the altered genes resulting from D2ghdh overexpression in well-differentiated myotubes (Fig. 6a). The downstream metabolites 2HG were decreased, which was consistent with the theoretical metabolic trend (Fig. 6b). Joint pathway analysis revealed several pathways, such as CoA biosynthesis and glutathione metabolism (Fig. 6c). To reveal the direct effect of D2ghdh overexpression on well-differentiated myotubes, GO analysis was used to depict the molecular function, biological process, and cellular component (Supplementary Fig. S8a–d). Microtubule motor activity, microtubule-binding, tubulin binding, fibronectin-binding, ATPase activity, cell adhesion molecule binding, and activin receptor binding were the top molecular function.

Next, we compared the transcriptional and metabolic characteristics between D2HG-treated myotubes overexpressing D2ghdh and NTC treated myotubes overexpressing D2ghdh to evaluate the D2hgdh overexpression on the alleviation of D2HG-induced proteolysis (Supplementary Data 11). It was found that 241 genes were up-regulated, and 306 genes were down-regulated after D2HG treatment (Fig. 6d). Lactate, 3-methylhistidine, carnitine, and 2-hydroxyglutarate were enriched, while NADPH and creatinine were depleted. GO analysis revealed the molecular function of oxidoreductase activity, acting on the CH – NH group of donors as the top pathway enrichment (Supplementary

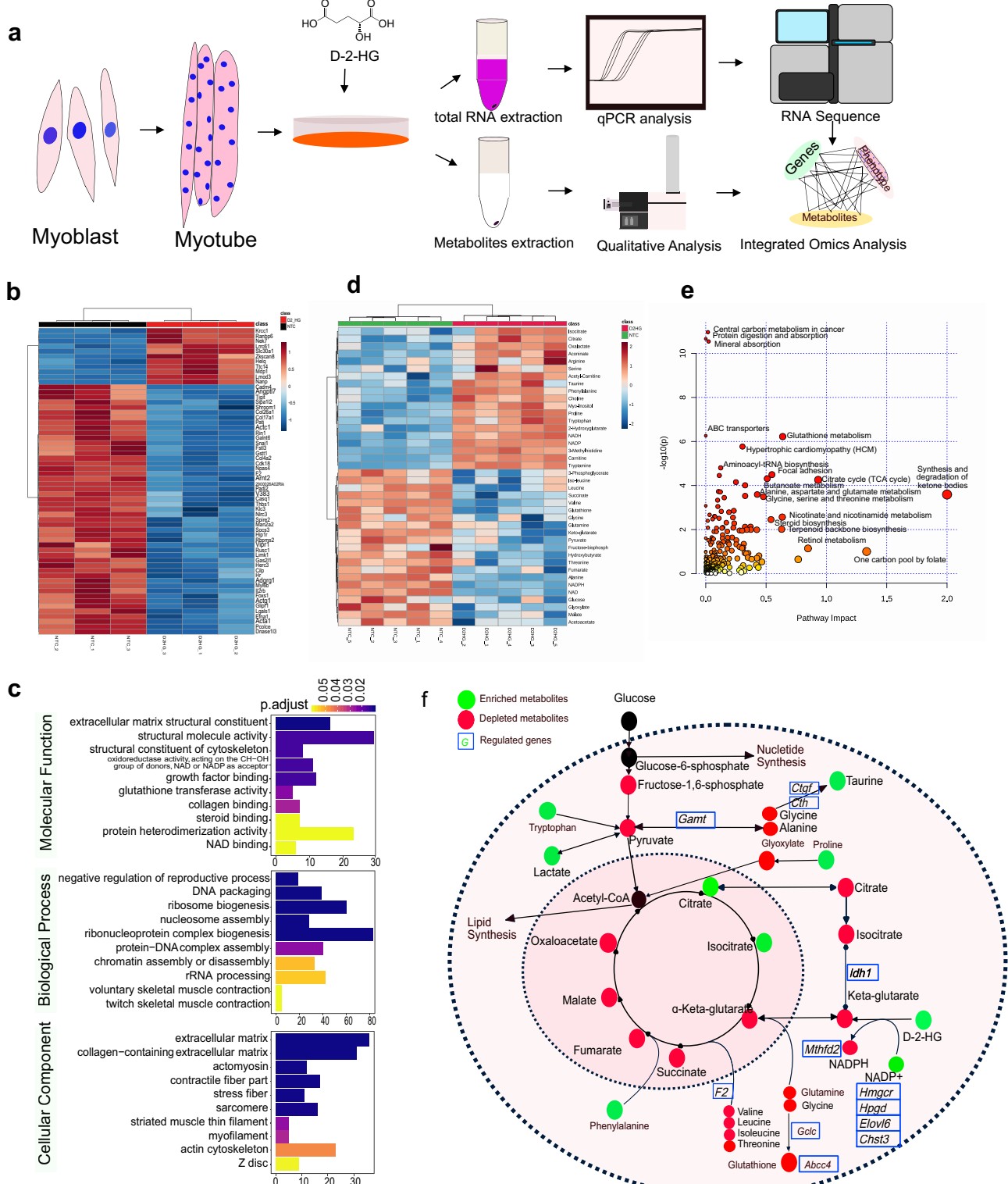

**Fig. 4 D2HG induces muscle wasting through activation of the UPP and metabolic reprogramming of the hydroxy compound biosynthesis. a** Schematic of the ex vivo experiment designed to determine whether D2HG directly induces muscle atrophy. **b** Heatmap of differentially expressed genes in D2HG-treated well-differentiated myotubes compared to NTC myotubes. All experiments were performed in triplicate. Gene expression values were expressed as the fragments per kilobase per million. **c** GO enrichment analysis of D2HG versus control well-differentiated myotubes. **d** Heatmap of the relative abundance of differential metabolites between D2HG-treated well-differentiated myotubes and NTC myotubes. Each group had five replicates. **e** Integrated analysis of transcriptome profiling and metabolomics profiling revealed the significantly altered pathway resulting from D2HG. **f** Summary figure highlighting metabolites on simplified metabolic pathways adapted from the KEGG metabolic pathways. Red circles indicate decreased metabolites; blue circles indicate increased metabolites.

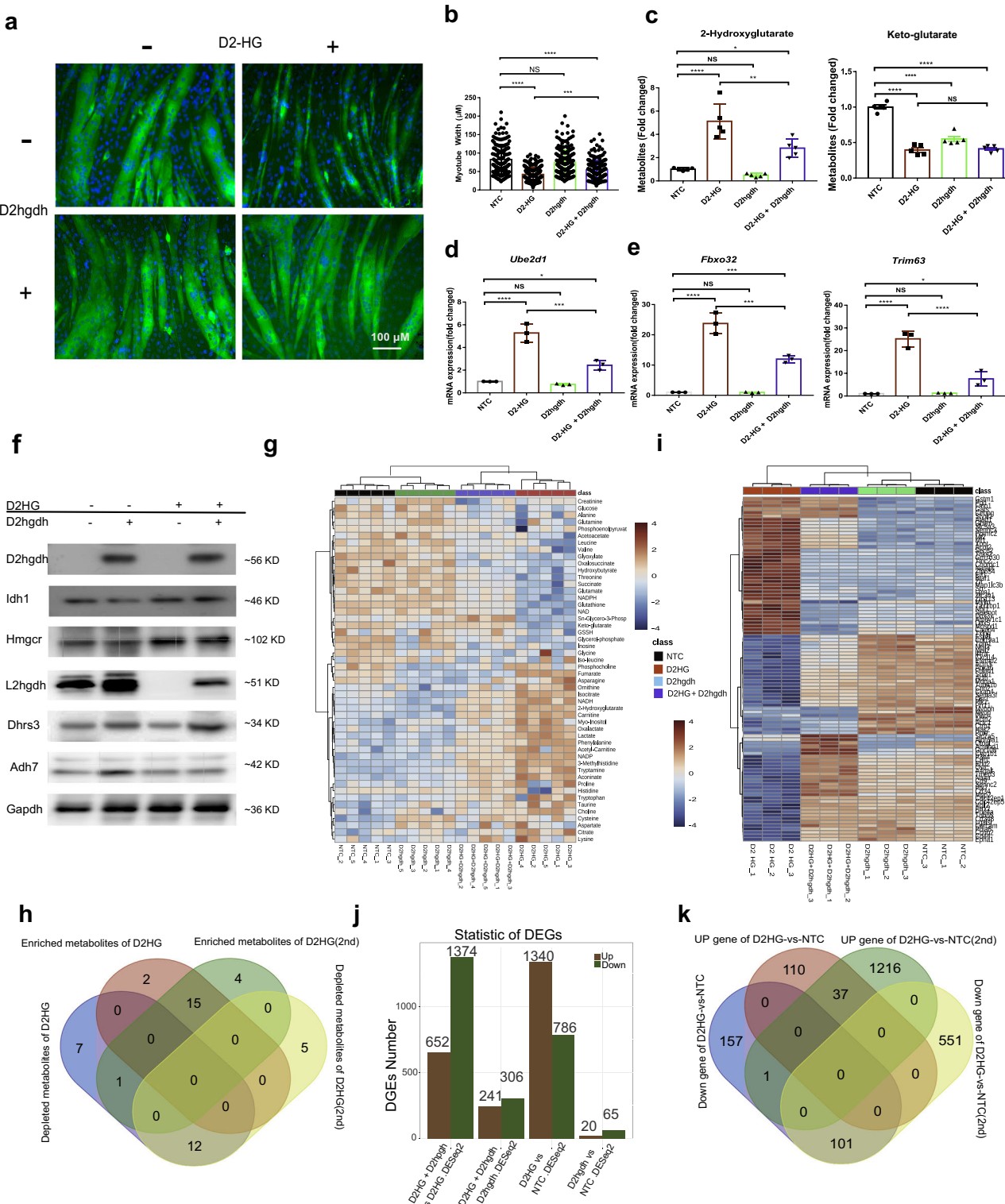

Fig. S9a–d). In addition, D2ghdh induced the alteration of biological processes, such as sterol biosynthetic process, organic hydroxy compound metabolic process, and acetyl-coA metabolic process.

Furthermore, D2HG-treated D2ghdh overexpressing myotubes and D2HG-treated control myotubes were also compared (Supplementary Data 12). A total of 1356 genes were depleted, and 640 genes were enriched (Fig. 6e). ATP-dependent serine/

threonine kinase regulator activity was the top molecular function (Supplementary Fig. S10a–d).

Gene Set Enrichment Analysis (GSEA) was used to interpret the gene expression data from D2hgdh overexpression and D2HG treatment. D2HG-treated well-differentiated myotubes overexpressing D2hgdh showed enriched proteasome accessory complex and SUMO transferase activity, as well as decreased positive regulation of stem cell differentiation (Fig. 6f). These functional

**Fig. 5 D2hgdh reversed D2HG-induced muscle wasting by inhibiting the UPP-mediated proteolysis and metabolic reprogramming. a** Typical immunofluorescence morphological changes of D2hgdh overexpressing and NTC myotubes after metabolite D2HG treatment (scale bar: 100 μm). **b** Column plot of myotube width in NTC and D2HG-treated well-differentiated myotubes with / without D2hgdh overexpression. One-way ANOVA with post hoc Tukey's multiple comparison test (****$p < 1e-4$, ***$p < 2e-4$, **$p < 2e-3$, *$p < 0.05$). **c** Column plot of metabolite changes in NTC and D2HG treated well differentiated myotubes with/without D2hgdh overexpression. One-way ANOVA with post hoc Tukey's multiple comparison test (****$p < 1e-4$, ***$p < 2e-4$, **$p < 2e-3$, *$p < 0.05$). **d**, **e** Column plot of mRNA expression of *Ube2d1*, *Trim63* and *Fbxo32* in NTC and D2HG-treateddwell-differentiated myotubes with/out D2hgdh overexpression. One-way ANOVA with post hoc Tukey's multiple comparison test (****$p < 1e-4$, ***$p < 2e-4$, **$p < 2e-3$, *$p < 0.05$). **f** Western blot results of protein expression in NTC and D2HG-treated well-differentiated myotubes with/without D2hgdh overexpression. GAPDH was used as an internal reference. **g** Heatmap of relative abundance of differential metabolites in NTC and D2HG-treated well-differentiated myotubes with/without D2hgdh overexpression. Each group had five replicates. **h** Venn diagram showing commonly altered metabolites resulting from D2HG treatment. **i** Heatmap of differentially expressed genes in NTC and D2HG-treated well-differentiated myotubes with/without D2hgdh overexpression. Each group had three replicates. **j** Summary of deferentially expressed genes in NTC and D2HG treated well-differentiated myotube with/without D2hgdh overexpression. **k** Venn diagram showing commonly altered genes as a result of D2HG treatment.

annotations confirmed the D2HG mediated enhanced proteolysis and attenuated muscle differentiation.

To show the commonality relationships, the altered genes (Supplementary Data 13) and Venn diagrams were used. Fifty-five common enriched genes and 52 common depleted genes were found (Fig. 6g). Metabolomics showed a number of common metabolic profiles, including depletion of 12 metabolites and enrichment of 8 metabolites after D2hgdh overexpression in well-differentiated myotubes, and depletion of 15 metabolites and enrichment of 12 metabolites after D2HG treated myotubes overexpressing D2hgdh when compared with D2HG treated control myotubes (Supplementary Data 14). Venn diagram showed a common enriched metabolite and 5 commonly depleted metabolites (Fig. 6h). Overexpression of D2hgdh resulted in decreased 2HG, as well as isocitrate, carnitine, Fum, and NADH. D2hgdh could catalyze the conversion of D2HG to keto-glutarate with the driver of NAD to NADH. It was interesting that D2HG treatment resulted in a decrease in an increase in the $NAD^+$/NADH ratio. However, when D2hgdh was overexpressed in myotubes, it could release the effect of D2HG and rescue the depletion of NADH by cycling the redox balances and metabolic homeostasis[31]. Excessive D2HG could induce dysfunction of well-differentiated myotubes, while D2hghd overexpression myotube could catalyze D2HG and take advantage of metabolic reprogramming to drive the cycle of redox balances and metabolic homeostasis (Fig. 6i).

**Ivosidenib relieves *IDH1* mutation mediated exacerbated cancer cachexia.** Ivosidenib is a selective inhibitor of *IDH1* mutation that blocks the abnormal *IDH1* protein and can reduce abnormal D2HG levels[32]. We evaluated the effect of ivosidenib on alleviating *IDH1* mutation-exacerbated cancer cachexia in CT26 tumor-bearing mice (Fig. 7a). From DPI 9 of palpable tumor, mice bearing CT26 tumor and mice bearing CT26 tumor with *IDH1* mutation were intravenously administered with 50 mg/kg ivosidenib or PBS as NTC control every day for the following experimental period[32]. Mice bearing CT26 tumor with *IDH1* mutation showed a worse survival compared to the control mice (Fig. 7b). After ivosidenib treatment, the survival of mice bearing CT26 tumor did not change significantly, while the survival of mice bearing CT26 tumor with an *IDH1* mutation was prolonged. As for the tumor weight curve, there were no changes between the two groups that did not receive treatment. However, after treatment, the tumor weight of control mice did not change significantly, while it was decreased in CT26 tumors with an *IDH1* mutation. Ivosidenib preserved lean body weight in the *IDH1* mutation tumor and delayed the progression of cancer cachexia, as cancer cachexia was first observed at DPI 25 (Fig. 7c). To verify that ivosetinib has a direct effect on cachexia directly, we compared the lean body

weight with a similar tumor weight of 1.56 g (tumor volume 3000mm³) and found that there was a significant preservation of lean body weight in ivosidenib treated mice bearing CT26 tumor with *IDH1*-mutation when compared to NTC treated mice bearing CT26 tumor with *IDH1*-mutation (Fig. 7d). Next, we measured the cross-sectional area of the muscle gastrocnemius (Fig. 7e), and found that the *IDH1* mutation group was smaller than the control group (Fig. 7f). Ivosidenib treatment prevented muscle wasting. The *IDH1* mutation also led to a decrease in muscle gastrocnemius weight, while ivosidenib reversed the loss of muscle gastrocnemius (Fig. 7g). The mRNA expression of *Trim63* and *Fbxo32* was consistent with the his-topathologic findings (Supplementary Fig. S11). *IDH1* mutation in CT26 tumor-bearing mice resulted in increased expression of *Ube2d1*, *Trim63*, and *Fbxo32*, while ivosidenib treatment inhibited the upregulation of UPP-related enzymes. Serum D2HG concentration was increased after *IDH1* mutation, whereas it was decreased after ivosidenib treatment (Fig. 7h). These results indicated that D2HG was a mediator of muscle wasting. Inhibition of the production of D2HG production by *IDH1* mutation inhibitor ivosidenib or catabolism of D2HG by overexpression of D2HGDH may reverse D2HG induced muscle proteolysis and slow the progression of cancer cechexia.

Since *IDH1* mutation was frequently found in glioma. We also established an orthotopic tumor model of in vivo bearing GL261 glioma cells with *IDH1*-mut in vivo. It was found that the import of mutant *IDH1* into the GL261 glioma tumor resulted in a significant improvement in survival (Fig. 7i). Ivosidenib could prolong the survival of mice bearing GL261 glioma tumor with mutant *IDH1*, but did not change the survival of mice bearing wild-type GL261 glioma tumor. Loss of body weight was observed and >5% body weight loss occurred at DPI 15 for the mice bearing wild-type glioma tumor and *IDH1* mutant glioma tumor (Fig. 7j). After ivosidenib treatment, the cachexia syndrome occurred at DPI 17 for the mice bearing wild-type glioma tumor, while at DPI 20 for the mice bearing *IDH1* mutant glioma tumor bearing mice. There was a significant decrease in muscle area (Supplementary Fig. S12a, b). Ivosidenib treatment resulted in significant improvement of muscle area. Although there was no significant change in muscle weight was found (Supplementary Fig. S12c), the expression of E3 ligases *Trim63* and *Fbxo32* was increased in the muscle of mice bearing *IDH1* mutant glioma tumors, whereas it was attenuated after ivosidenib treatment (Supplementary Fig. S12d). Protein expression of the E3 ligases Atrogin-1 and MuRF1 was also improved by ivosidenib treatment (Supplementary Fig. S13a). The protein degradation marker of serum 3-methylhistidine was increased in the mutant *IDH1* tumor-bearing mice, whereas it was decreased after ivosidenib treatment (Supplementary Fig. S13b). The

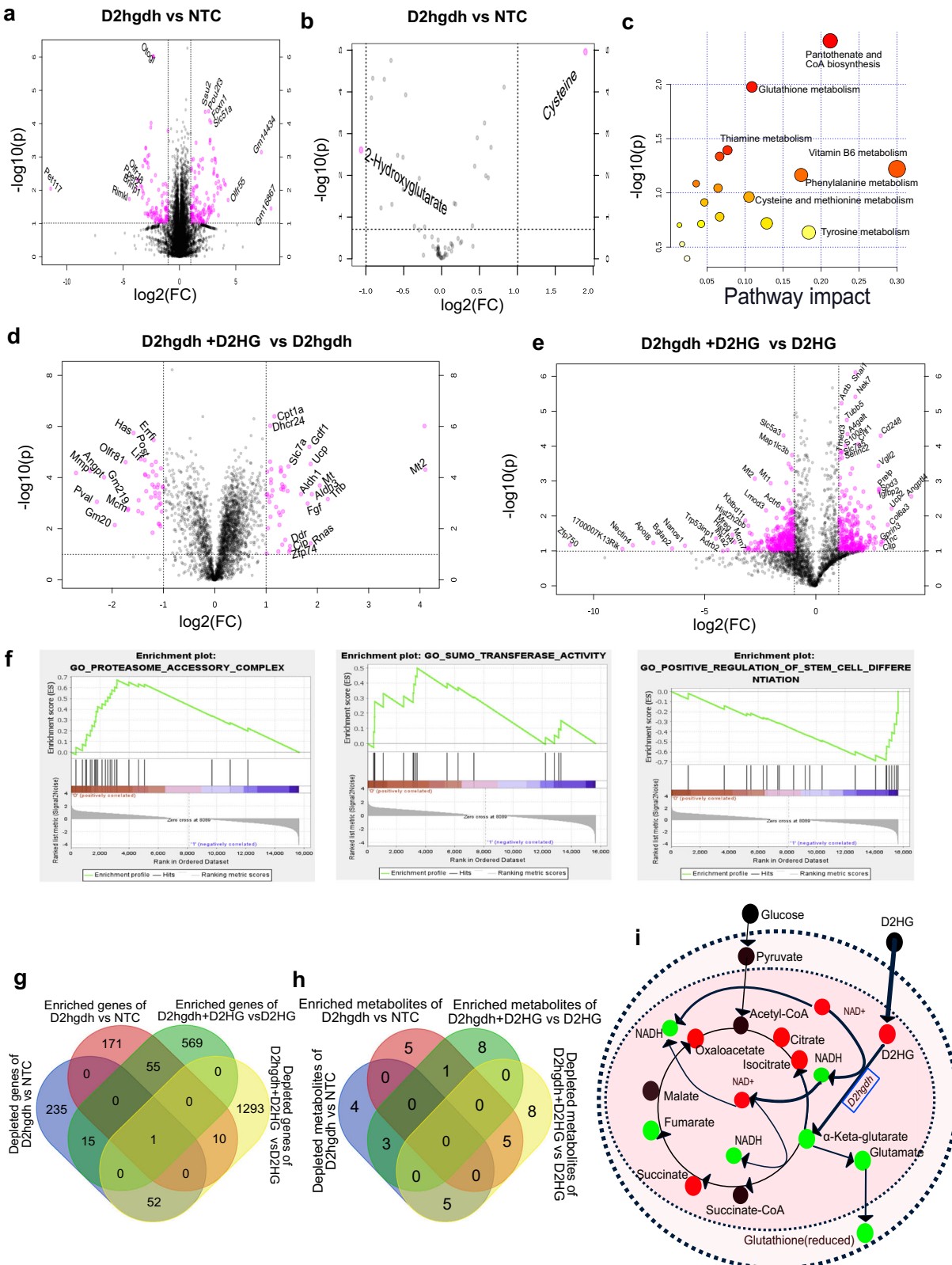

phosphorylation levels of the key protein synthesis pathway mTOR and p70s6k were increased, while their downstream 4E-BP1 phosphorylation levels were decreased (Supplementary Fig. S13c). In addition, serum D2HG concentration was increased after *IDH1* mutation, whereas it was decreased after ivosidenib treatment (Fig. 7k).

## Discussion

For well-differentiated skeletal muscle, cancer induces a catabolic proteolysis response characterized by the activation of protein degradation pathways, mainly through the UPP[3]. The present study showed that the oncometabolite D2HG induces protein degradation and muscle atrophy by increasing protein

**Fig. 6 D2hgdh reversed D2HG-induced muscle wasting through metabolic reprogramming. a** Volcano plot showing the significantly altered genes between the D2hgdh overexpression and NTC groups. Significance was defined as fold change 2.0 and $p < 0.05$. **b** Volcano plot showing the significantly altered metabolites between the D2hgdh overexpression and NTC groups. Significance was defined as fold change 2.0 and $p < 0.05$. **c** Integrated analysis of transcriptome profiling and metabolomics profiling showing the significantly altered pathway resulting from D2hgdh overexpression. **d** Volcano plot showing the significantly altered genes between D2HG-treated D2hgdh overexpression myotubes and negatively treated D2hgdh overexpression myotubes. Significance was defined as fold change 2.0 and $p < 0.05$. **e** Volcano plot showing the altered genes between D2HG-treated D2hgdh overexpressing myotubes and D2HG-treated control myotubes. Significance was defined as fold change 2.0 and $p < 0.05$. **f** Gene Set Enrichment Analysis of altered signaling pathway resulting from D2HG. **g** Venn diagram showing commonly altered genes resulting from D2hgdh overexpression. **h** Venn diagram showing the commonly altered metabolites resulting from over-expression. **i** Summary plot of metabolites along a simplified metabolic pathway adapted from KEGG metabolic pathways. Red circles indicate decreased metabolites; blue circles indicate increased metabolites.

degradation and decreasing protein synthesis. Based on the in vitro experiment, D2HG treatment increased the expression of UPP E3 ligase genes, decreased myotube diameter, inhibited differentiation, and induced distinct transcriptional and metabolic features. This was consistent with previous reports that D2HG was associated with myopathy[12,20]. Furthermore, the in vivo experiment showed that the *IDH1* mutation in CT26 cancer resulted in high serum levels of D2HG, loss of lean body weight and skeletal muscle. Increasing gene and protein expression of the E3 ligases MuRF1 (*Trim63*) and Atrogin-1 (*Fbxo32*) reflects enhanced protein degradation. Whereas decreasing protein expression of phosphorylated levels of mTOR, P70S6K and 4E-BP1 indicates attenuated protein synthesis. The overall metabolic mechanism of *IDH1* mediated D2HG accumulation and its contribution to protein turnover and muscle wasting is showed in Fig. 8.

The oncometabolite D2HG can accumulate up to millimolar levels in cancer patients[6]. The imbalance between D2HG formation and degradation has been associated with cancer specific metabolism, redox homeostasis, immunosuppressive, genetic and epigenetic effects[33]. In this study, high D2HG was found in the cancer cachexia group. In addition, patients with *IDH1* mutation had high levels of D2HG levels. Based on TCGA datasets, patients with *IDH1* mutation had poor survival. Wild-type *IDH1* catalyzes the oxidative decarboxylation of isocitrate to α-ketoglutarate with the concomitant production of NADPH. Mutations in *IDH1* have been reported in various types of cancer, particularly in low-grade gliomas[18,19], cholangiocarcinoma[14], lung cancer[17], renal cancer[6], pancreatic adenocarcinoma[15], and gastrointestinal cancer[11]. However, IDH mutations have a different prognostic value depending on the type of cancer. In gliomas, *IDH1* mutation explains the adverse prognostic effect of older age[18,19], while in intrahepatic cholangiocarcinoma patients with mutant IDH, there was no direct association with clinical outcome[34]. In general, these solid tumors have a high incidence of cancer cachexia syndrome[3]. In this study, we cloned the cDNA of *IDH1*-R132H under the promoter of EF1α and produced a lentivirus to gain the function of *IDH1*-R132H in CT26 colon cancer cells and GL261 glioma cells. When the CT26 tumor cells with *IDH1*-R132H were transplanted into BALB/c mice, the overall survival was shorter, and the lean body weight decreased faster. In addition, the cachexia occurred earlier in the presence of the *IDH1* mutation. However, when the GL261 glioma cells with *IDH1*-R132H were transplanted into C57BL/6 J mice, the survival was improved. There was no significant difference in body weight loss, and the incidence of cachexia was similar. The *IDH1* inhibitor ivosidenib was able to significantly improve survival.

The mechanism of cancer-related loss of skeletal muscle mass is complex[1,3]. Many mediators, including inflammatory cytokines, chemokines, glucocorticoids, and metabolites, are elevated as a result of host-cancer interactions[1]. These mediators are released and delivered to the skeletal muscle through the circulatory system. When these mediators act on skeletal muscle, they disrupt substrate metabolism and cause functional changes. UPP activation has been observed in the wasting muscles from several in vitro and in vivo experiments and has been recognized as a key mechanism for muscle atrophy[3]. In the present study, D2HG treatment in well-differentiated myotubes directly resulted in short myotube diameter and upregulated E3 ligases and E2 ubiquitin-conjugating enzyme. The gain of the function of *IDH1* mutation in the CT26 tumor also resulted in high circulating D2HG levels and upregulated E3 ligases in skeletal muscle. A GSEA of the transcriptional features revealed that D2HG treatment in well-differentiated myotubes resulted in enrichment of the proteasome accessory complex and SUMO transferase activity and depleted positive regulation of stem cell differentiation. This study found that *Sema3c, Hoxb4, Tgfb2, Gata6, Nkx2-5, Ltbp3, Sox5*, and *Ptn* were depleted by D2HG. *Gata6* is required for cardiovascular development and myosin heavy chain gene expression[35]. *Nkx2-5* has RNA toxicity in myotonic muscular dystrophy[36]. Ectopic expression of *Ltbp-3* in mature mouse skeletal muscle increases fiber area and decreases myostatin activity[37]. *Ptn* is a heparin-binding growth factor that impairs muscle reinnervation and reduces cell density in muscle gastrocnemius injury[38]. Therefore, these factors may be mediators of the effects of D2HG. On the other hand, D2ghdh overexpression in D2HG-treated myotubes affected the extracellular matrix, myosin complex, and spindle microtubule. Myosin is a motor protein involved in the generation of mechanical force in myofibers. Muscle myosin molecules are heterohexamers composed of two myosin heavy chains and two myosin light chains[25]. Selective loss of myosin contributed mainly to skeletal muscle wasting during cancer cachexia via the UPP, confirming that the high levels of D2HG induced muscle atrophy via UPP-mediated proteolysis.

The integrated analysis of transcriptome and metabolome revealed the distinct effect of D2HG on well-differentiated myotubes through metabolic reprogramming. Accumulated D2HG impaired the myotube function and induced proteolysis. D2HG increased the $NAD^+$ / NADH ratio. While D2hgdh overexpression in myotubes alleviated D2HG induced an increase in the $NAD^+$/NADH ratio. D2hgdh is a mitochondrial enzyme that catalyzes the conversion of D2HG to keto-glutarate with the driver of NAD+ to NADH. When D2hgdh was overexpressed in myoblasts, the differentiation from myoblast to myotube was unaffected. The rescue of protein proteolysis after treatment with 2HG in the D2hgdh overexpressing myotubes indicated the reprogramming of redox homeostasis.

Metabolic alterations in cancer have been recognized as a potential therapeutic target. In this study, we inhibited the production of D2HG in *IDH1* mutant tumor using the selective *IDH1* inhibitor ivosidenib. Our results showed that ivosidenib could delay the progression of cancer cachexia syndrome. Furthermore, ivosidenib preserved skeletal muscle mass by decreasing the expression of the E3 ligases *Trim63* and *Fbxo32*, as well as inhibiting the production of D2HG in tumor and serum.

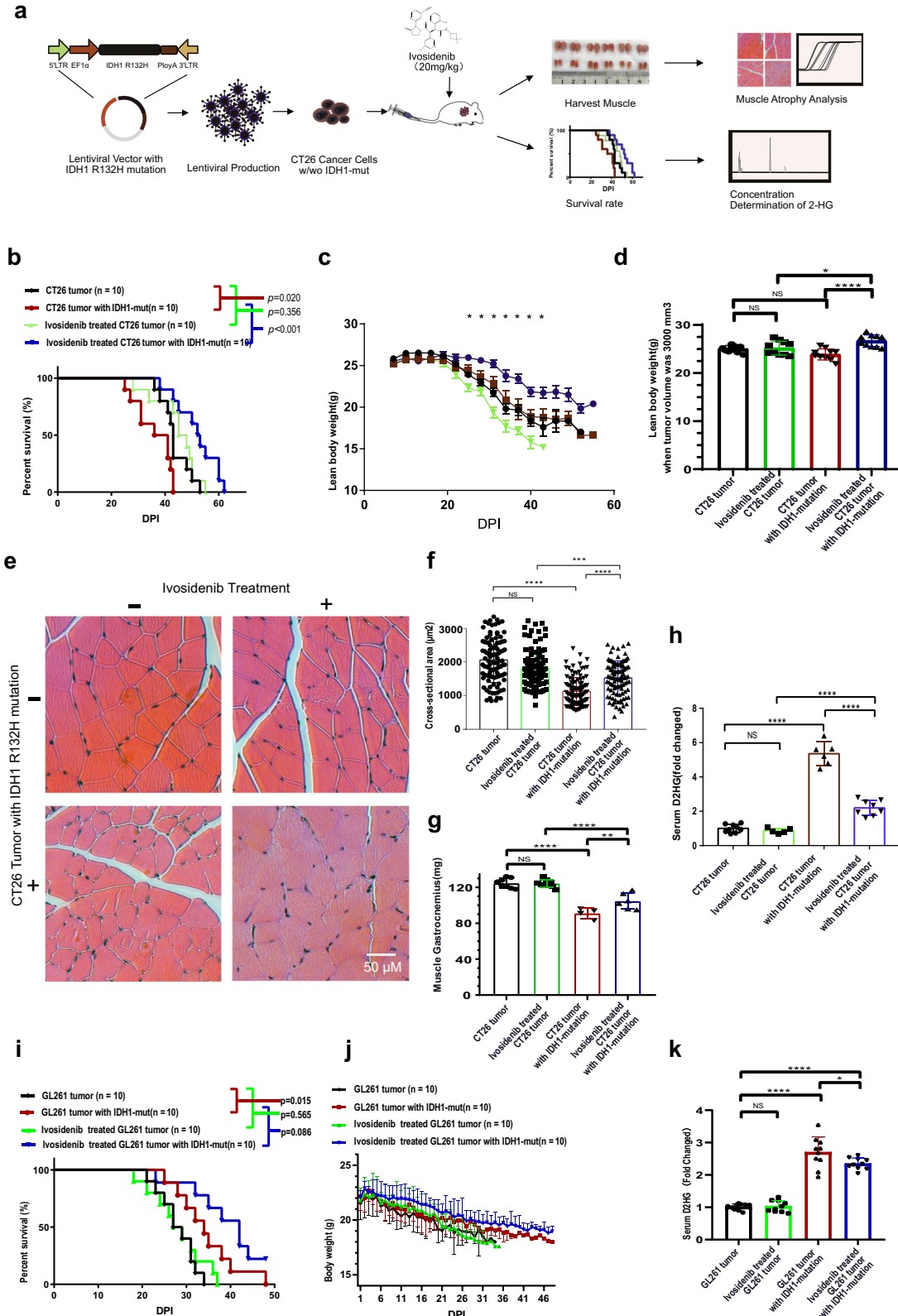

Cancer cachexia is characterized by loss of lean body weight and wasting of skeletal muscle. Refractory cachexia is a contra-indication to many treatments and clinical trials. Although several mediators of cachexia have been studied[1], genetic factors in the progression of cancer cachexia have been neglected. In general, gene mutation or differential expression has not been recognized as a direct responsible factor responsible for proteolysis and cachexia symptoms. In the present study, cancer patients with *IDH1* mutation had high levels of D2HG, which could exacerbate proteolysis and induce cancer cachexia. These results highlight the importance of precision medicine for cancer cachexia and the comprehensive treatment of cancer patients.

Several limitations warrant further investigation. There are extensive metabolic abnormalities in cancer patients, and

**Fig. 7 Ivosidenib ameliorates *IDH1* mutation-mediated skeletal muscle atrophy and inhibits serum D2HG accumulation. a** Schematic of the in vivo experiment. **b** The survival curve of the *IDH1* (R132H) mutation cancer and wild-type cancer groups treated with ivosidenib or NTC. Log-rank (Mantel-Cox) test. **c** The lean body weight of mice in the *IDH1* (R132H) mutant cancer group and wild-type cancer group treated with ivosidenib or NTC. ANOVA followed by Tukey's multiple comparison test; the lean body weight difference between ivosidenib treated mice bearing *IDH1* mutant tumor vs. NTC treated mice bearing *IDH1* mutant tumor was significant (*$p < 0.05$). **d** Mice with a similar tumor weight of 3000 mm^3 (1.56 g), the lean body weight was compared in *IDH1*-R132H and *IDH1*wt cancer treated with ivosidenib or NTC. One-way ANOVA with post hoc Tukey's multiple comparison test (****$p < 1e-4$, ***$p < 2e-4$, **$p < 2e-3$, *$p < 0.05$). **e** Typical cross sectional histopathological image of muscle gastrocnemius in *IDH1* (R132H) mutant cancer and wild-type cancer treated with ivosidenib or NTC (scale bar: 50 μm). **f** Column plot of the cross sectional area of muscle gastrocnemius in *IDH1* (R132H) mutation cancer and wild-type cancer treated with ivosidenib or NTC. One-way ANOVA with post hoc Tukey's multiple comparison test (****$p < 1e-4$, ***$p < 2e-4$, **$p < 2e-3$, *$p < 0.05$). **g** Column plot of muscle gastrocnemius weight in *IDH1*-R132H and *IDH1*wt cancer treated with ivosidenib or NTC. One-way ANOVA with post hoc Tukey's multiple comparison test (****$p < 1e-4$, ***$p < 2e-4$, **$p < 2e-3$, *$p < 0.05$). **h** Column plot of serum D2HG concentration in the *IDH1* (R132H) mutant cancer group and the wild-type cancer group treated with ivosidenib or NTC. One-way ANOVA with post hoc Tukey's multiple comparison test (****$p < 1e-4$, ***$p < 2e-4$, **$p < 2e-3$, *$p < 0.05$). **i** The survival curve of *IDH1* (R132H) mutation cancer and wild-type cancer groups treated with ivosidenib or NTC. Log-rank (Mantel-Cox) test. **j** The body weight of mice in the *IDH1* (R132H) mutant GL261 tumor group and wild-type GL261 tumor group treated with ivosidenib or NTC. **k** Column plot of serum D2HG concentration in *IDH1* (R132H) mutant GL261 cancer group and wild-type GL261 cancer group treated with ivosidenib or NTC. One-way ANOVA with post hoc Tukey's multiple comparison test (****$p < 1e-4$, ***$p < 2e-4$, **$p < 2e-3$, *$p < 0.05$).

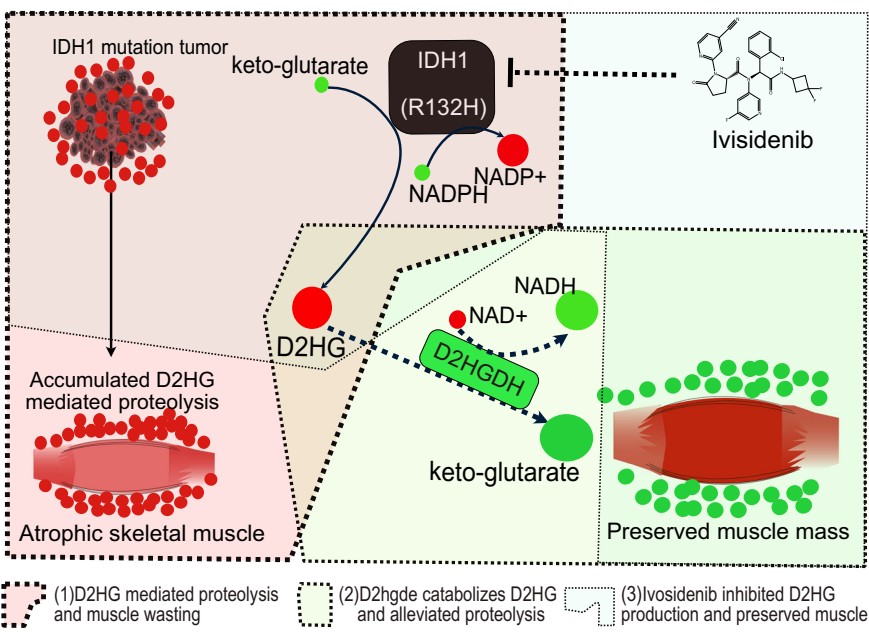

**Fig. 8 The metabolic pathway of *IDH1* mediated D2HG accumulation and its contribution to proteolysis and muscle wasting.** The accumulated D2HG can be catabolized by the enzyme D2HGDH and the production of D2HG can be inhibited by the *IDH1* inhibitor ivosidenib.

different types of cancer have different metabolic profiles. Our experiment provided evidence that pathological concentration of D2HG induced proteolysis and muscle atrophy. The contribution of other altered cancer metabolites to muscle metabolism and atrophy, such as succinate, fumarate, D2HG, 1-methylhistidine, 3-methylhistidine, and 4-hydroxyproline, at physiological and pathological concentrations, is worthy of attention. Another limitation of this study was the genetic contribution to the development of cancer cachexia. *IDH1* mutation resulted in significantly shorter survival time in subcutaneous tumor model with CT26 colon cancer tumor, and significantly longer survival time in orthotopic tumor model with GL261 glioma tumor. However, the treatment of ivosidenib in both models showed prolonged survival and delayed weight loss in both models. After analyzing the pharmacodynamic characteristics, we concluded the direct contribution of oncometabolite D2HG on proteolysis and muscle wasting. Thirdly, there was a bias in the endpoint of the murine cancer cachexia model because mice were individually culled when reaching the humane endpoint that the animal was declared sick by our veterinarian. When we obtained the

significantly different experimental endpoint of change in lean body weight, all live mice were euthanized. Future studies are needed to further investigate the potential of ivosidenib for individualized anti-cachexia effect in cancer patients with an *IDH1* mutation. Fourthly, growing studies in humans and rodent models showed sex differences in normal skeletal muscle and cancer cachexia[39]. Male cancer patients generally have a higher prevalence of cachexia, greater weight loss or muscle wasting, and worse outcomes than female cancer patients[40]. We thus used male mice bearing CT26 colon carcinoma cells and GL261 glioma cells for the cancer cachexia model as most studies have done[41–44]. Further studies to evaluate the sex differences in *IDH1* mutation and cancer cachexia progression would improve the basic mechanisms and treatment strategy[45].

In conclusion, mutant *IDH1* mediated D2HG accumulation leads to metabolic reprogramming and distinct transcriptional features in differentiated myotubes. Catabolism of D2HG by the overexpression of D2hghd and inhibition of D2HG production by the *IDH1* mutation inhibitor ivosidenib could reverse D2HG-mediated proteolysis and muscle atrophy. These results

demonstrate the potential for personalized treatment of cancer cachexia in patients with an *IDH1* mutation.

## Materials and methods

**Study approval and sample collection**. Ethical approval was obtained from the Health Research Ethics Board of Shanghai Jiao Tong University Affiliated Sixth People's Hospital. Patients were recruited between July 2013 and May 2020 (Registration number: ChiCTR-DDD-17013590 at www.chictr.org.cn). Written informed consent was obtained from all participants. All ethical regulations relevant to human research participants were followed. Sample identification numbers were used as unique, anonymous identifiers that were independent of each patient's true identifiers. Cancer patients with and without cachexia were recruited based on international consensus diagnostic criteria[2]. Cancer patients with cachexia were included according to the following criteria: weight loss > 5% in the last 6 months, or weight loss >2% in the last 6 months, and a body mass index (BMI) <20 kg/m[2]. Weight-stable cancer patients were those with a BMI <25 kg/m[2] but no significant weight change in the previous year. Age, height, weight, cancer biomarkers, and biochemical biomarkers were obtained from patients' laboratory reports, either from either the date of diagnosis or the date closest to diagnosis. To exclude the effect of chemotherapy on the production of D2HG, the cachectic and weight-stable cancer patients were free of chemotherapy for at least 21 days at the time of samples collection. Exclusion criteria were: renal or hepatic failure; acquired immunodeficiency syndrome; inflammatory bowel disease; systemic infection.

**Tissue genotyping**. The *IDH1* genotype was determined by Sanger sequencing. Formalin-fixed and paraffin-embedded tumor samples were retrieved to extract DNA using QIAamp DSP DNA FFPE Tissue Kit (Qiagen, Venlo, Netherlands). PCR amplification was performed in a 20-µl reaction mixture containing 200 ng of DNA, 0.6 µL of forward and reverse primers, 10 µl of 2× PCR Master Mix buffer (Qiagen, Venlo, The Netherlands) and RNase-free water. PCR reaction was performed at 96 °C for 30 s, followed by 35 cycles of denaturation at 96 °C for 10 s, annealing at 55 °C for 10 s, and extension at 72 °C for 20 s. The products were sequenced and the results were aligned to identify the mutation of *IDH1*.

**Serum D2HG determination**. D2HG levels in sera were determined using a 6490 Triple Quadrupole LC/MS[12]. Briefly, 200 µL of thawed serum was mixed with 800 µL of acetonitrile and centrifuged at 10,000 rpm for 10 min at room temperature (24 °C). The supernatants were then transferred to polypropylene tubes and evaporated under a vacuum. The residue was reconstituted with 50 µL of 90% methanol in water (v/v), and 10 µL was injected into an Agilent 6490 Triple Quadrupole LC/MS system. Chromatographic separation was performed on a Chirex 3126 d-penicillamine, LC column (150 × 4.6 mm, Phenomenex, Torrance, CA, USA). Data acquisition, peak integration, and processing were performed using MassHunter software (Agilent Technologies, Santa Clara, CA, USA). MS-grade reagents were used for extractions. Pure D2HG was used as standard for the calculation of D2HG levels.

**Cell culture**. All cell lines were verified to be mycoplasma-free using cell culture supernatant. The CT26.wt colon carcinoma cells (ATCC; USA) and GL261 glioma cells (Cell Bank of the Typical Culture Preservation Committee of the Chinese Academy of Sciences, China) were cultured in Dulbecco's Modified Eagle Medium (DMEM) supplemented with 10% fetal bovine serum

and penicillin-streptomycin (100 U/mL and 100 mg/mL, respectively) in a humidified atmosphere containing 5%$CO_2$/95% air at 37 °C.

Murine C2C12 myoblasts were obtained from the cell bank of the Typical Culture Collection Committee of the Chinese Academy of Sciences. Low-passage C2C12 myoblasts were first cultured in growth DMEM medium supplemented with 10% fetal bovine serum and penicillin-streptomycin (100 U/mL and 100 mg/mL, respectively) in a humidified atmosphere containing 5%$CO_2$ air at 37 °C. C2C12 myoblasts (<15 passages) were induced by replacing the culture medium with fusion medium (DMEM containing 2% horse serum and 1% penicillin/streptomycin). When the C2C12 myoblasts reached ~80–90% confluence in the growth medium, a substantial volume of the medium was added, and incubation was continued for 4 days. The differentiation medium was changed every 3–4 days after myotube formation were formed. Cell culture based experiments were performed in independent triplicates.

**Ex vivo myotube wasting model and metabolites treatment**. For the in vitro myotube wasting model, well-differentiated myotubes exposed to 100 µM dexamethasone were positively controlled[46]. Metabolites were added to the differentiation medium to induce muscle atrophy. These media were used for the culture of the myotubes for >2 days.

**Immunofluorescence**. Well differentiated myotubes from different groups were fixed with 4% formaldehyde for 20 min at room temperature. The myotubes were washed with PBST and blocked with 5% bovine serum albumin (dissolved in PBST) for 60 min at 37 °C. The cells were incubated with primary anti-myosin heavy chain antibody (1:200) overnight at 4 °C. The cells were washed three times with PBST and incubated with the appropriate Alexa Fluor 488-conjugated secondary antibody for 60 min at 37 °C, and then washed three times with PBST for 5 min. The nuclei were stained with DAPI for 3 min, followed by three washes with PBST. An anti-fluorescence quencher was then added, and fluorescence images were taken using a fluorescence microscope. IgG controls were used to validate the specificity of the myosin heavy chain primary antibody by only fluorescent secondary antibody staining. The width of each individual myotube was measured at the widest cross section and each myotube was measured in three different regions (the average value was the width). Each group contained 3 biological duplicates and ~30 individual myotubes.

**Overexpression of D2hgdh in C2C12 myoblast using a lentivirus**. The cDNA of *IDH1* with the R132H mutation (NM_005896.2) was amplified by PCR (primers are shown in Supplementary Data 15). The plasmid pLV-EF1α-FLAG-IRES-Puro (Plasmid #85132 from Addgene) was digested with EcoR1 and BamH1. The lentiviral plasmid pLV-EF1α-*IDH1*-R132H-FLAG-IRES-Puro was produced by Gibson assembly. The resulting plasmid was sequenced to verify the correct insertion of the insert. The D2hgdh overexpression lentivirus pLV-EF1α-D2hgdh-FLAG-IRES-Puro was also cloned. The lentiviruses were produced in low-passage HEK293 cells (Cell Bank of the Typical Culture Preservation Committee of the Chinese Academy of Sciences, China). Approximately 2 h before transfection, the complete DMEM medium was replaced by 13 mL of pre-warmed Opti-MEM medium (Invitrogen Inc., Carlsbad, CA, USA). For each 15-cm plate, 450 µL of Opti-MEM was mixed with 20 µg lentiviral transfer plasmid, 15 µg psPAX2 packaging plasmid, 10 µg pMD2.G envelope plasmid, and 130 µl polyethylenimine. The mixture was incubated for 15 min at room temperature and added to the cells. After 6–12 h, the Opti-MEM medium was

replaced with 20 mL of pre-warmed complete DMEM medium. Viral supernatant was collected at 48 h and 72 h post-transfection. Cell debris was removed by centrifugation (5 min at 2000 rpm). Lentiviruses were concentrated using AmiconUltra 100 kD ultrafiltration centrifuge tube (Millipore) and stored at −80 °C. To generate D2hgdh-overexpressing C2C12 myoblasts, 0.2 ml of concentrated lentivirus was added to low-passage C2C12 myoblasts at $5 \times 10^4$ cells per well in 24-well plates. The myoblasts were cultured for 16 h. The D2hgdh overexpressing myoblasts were selected with 1 μg/ml antibiotics for 7 days.

**Western blot**. Well differentiated myotubes were homogenised and solubilised in ice-cold RIPA buffer containing protease inhibitors. After incubation on ice for 30 min, the protein supernatant was collected after centrifugation at $12,000 \times g$ for 15 min at 4 °C. The protein concentration was measured using a BCA protein Assay Kit BCA assay kit (Thermo Fisher Scientific, Waltham, MA, USA). Equal amounts of protein were mixed with $5 \times$ loading buffer and boiled for 5 min. Then, 20 mg of protein was loaded onto a 10% SDS-polyacrylamide. Proteins were separated and transferred to PVDF membranes. The membranes were blocked for 1 h at room temperature with 3% bovine serum albumin in TBST buffer. The membranes were then incubated with primary antibodies (diluted with 3% bovine serum albumin in TBST buffer at 1:1000) at 4 °C overnight and then washed three times with TBST. The membranes were then incubated with diluted secondary antibody (horseradish peroxidase (HRP)-conjugated secondary antibodies diluted 1:3000 in TBST buffer) for 1 h at room temperature and washed three times with TBST. Protein expression was detected using a chemiluminescence imaging system, and quantified using the ImageJ program. Equal distribution of protein loading was verified by probing the blots with an anti-GAPDH antibody.

**Total RNA extraction and quantitative PCR**. Total RNA was extracted from various tissues using Trizol. First-strand cDNA from total RNA preparations was synthesised using M-Mlv reverse transcriptase after normalisation with nuclease-free water. Quantitative qPCR was performed by using SYBR Premix Ex Taq II and designated probes of interest genes with synthetic primers (Supplementary Data 15). All samples were run in triplicate. GAPDH was used as an internal positive control. Relative mRNA calculations were analysed using the $2^{-\Delta\Delta CT}$ method.

**mRNA seq**. Total RNA from each group was used for the mRNA library preparation following a standard RNA extraction protocol. After first-strand reaction buffer and random primer mix, 1000 ng of total RNA was fragmented and processed for first-strand cDNA synthesis and second-strand cDNA synthesis. The double-stranded cDNA was purified, and adaptor ligation was performed. The samples were then multiplexed using barcoded primers. The PCR conditions were one cycle of 30 s at 98 °C, followed by 11 cycles of 10 s at 98 °C, 75 s at 65 °C, and a final incubation of 5 min incubation at 65 °C. The adaptor ligated library was purified and sent for quality assessment. If the library was of good quality, the libraries were sequenced using Novaseq systems (Illumina, Inc., San Diego, CA, USA). Raw sequencing files were available via SRA with metadata from mRNA-seq. Each sample was performed in triplicate. The RNA-seq data were deposited in the NCBI Sequence Read Archive under accession number: SUB8925446 and SUB8927518.

**mRNA-seq data analysis**. Raw FASTQ files from RNA sequencing were analysed for transcription quantification using QianTang Biotech Co., Ltd (Suzhou, China). Transcriptome references were

obtained from Ensembl. To estimate transcript abundances, HISAT was applied to the aligned reads and summarized transcript abundances into gene-level expression levels. StringTie was used for transcript prediction, and Bowtie2 was used to align sequencing reads to long reference sequences. Packages of DEseq2, EBseq, NOIseq, and PossionDis packages were used to identify up- and down-regulated genes. KEGG and GO analyses were performed using Cluster Profiler. GSEA was performed using the Java application from the Broad Institute. The full gene set from the differential gene expression analysis was ranked by "beta" value and then used as an input for GSEA pre-ranked analysis with database reference C5 Gene Ontology - Biological Process (GO-BP). Visualisation, including volcano plots, bar charts, and Venn diagrams, was performed using the standard R packages ggplot2. Enriched and depleted genes from the differential gene expression analysis were defined with an adjusted *p*-value cut-off of 0.05 and a fold change of 2.0.

**Metabolomics and data analysis**. Each sample was tested in five duplicates, each containing 2 million live cells. To extract the intracellular metabolites, the cells were washed twice with PBS and resuspended in 800 μL of 80% (vol/vol) methanol (pre-cooled to −80 °C) on dry ice for 30 min. Tubes were incubated on ice for 20 min and centrifuged at 15,000 g for 15 min at 4 °C. The supernatant containing metabolites was then transferred to a new tube on dry ice and centrifuged again (see above). The mixture was then dried with Speedvac. Metabolites were dissolved with 80% (vol/vol) methanol, centrifuged at $18,000 \times g$, and stored at −80 °C until LC-MS analysis on an Agilent 6550 Q-TOF LC/MS system. Targeted metabolomics was analysed on an Agilent 6490 triple quadrupole LC/MS system. Liquid chromatography was optimised using the Kinetex 2.6-μm PS C18, LC column $150 \times 2.1$ mm (Phenomenex, Torrance, CA, USA). The eluents were A: 0.01 % formic acid in HPLC grade water and B: 0.01 % formic acid in acetonitrile. The gradient was set as follows: 0–2 min 5% B; 2–36 min 5–100% B; 36–40 min hold at 100% B and then returned to initial conditions for 2 min for column equilibration. The flow rate was set as 0.3 mL per minute.

Multiple reaction monitoring was used for the qualitative and quantitative analysis of purified standards (Sigma, St. Louis, MO, USA). The features of the spectra were extracted using Agilent Mass Hunter Qualitative Analysis software (version B 6.0.633.0). Each peak was checked, and the abundances of all metabolites were exported. Retention times of the standards were confirmed (Supplementary Data 16). Three normalization procedures, such as normalization by sum, log transformation, and auto-scaling, were used to compare individual features. The distance measure was set to Euclidean, and the clustering algorithm to Ward. Finally, the metabolic flowchart functions were constructed using Pathvisio v3.3.0 based on the KEGG database. The integrated analysis of the altered metabolites and genes was performed with the Joint Pathway analysis module of MetaboAnalyst 5.0. Volcano plots were used to filter metabolites of interest with significant fold changes at 2.0 and statistical significance at 0.05 using the software MetaboAnalyst 5.0.

**Multi-omics analysis**. The differential expressed genes from the RNA sequencing analysis and differential represented metabolites were entered into the common pathway analysis using the MetaboAnalyst Portal. Default parameters were used, with hypergeometric test for enrichment analysis, degree centrality for topology analysis, and gene-metabolite pathways for pathway databases. Pathways were considered statistically significant when *p*-values were <0.05.

**Cancer cachexia model of in vivo bearing CT26 cells with IDH1-mut**. All animal studies were approved by the Institutional Animal Welfare Committee at Shanghai Jiao Tong University Affiliated Sixth People Hospital in according with the government guidelines for animal manipulation in China (No.: 2021-0361). BALB/c mice bearing a CT26 colon cancer tumour was a commonly used mouse model used to study cancer cachexia in vivo[47]. Male patients and mice showed a higher prevalence of cachexia, greater weight loss and muscle wasting compared to female cancer patients[39,48], so we used male mice for the following in vivo experiment. The cancer cachexia model bearing CT26 tumor with IDH1 mutation was established by generating of a positive clone of CT26 cells expressing mutant IDH1. Briefly, we cloned IDH1-R132H into the lentivirus plasmid pLV-EF1α-FLAG-IRES-Puro and produced pLV-EF1α-IDH1-R132H-FLAG-IRES-Puro lentiviruses, which were used to infect CT26 colon adenocarcinoma cells. After selection by adding antibiotics, we obtained a positive clone of CT26 cells with mutant IDH1 (CT26-IDH1-mut cells). We then checked the protein and mRNA expression of mutant IDH1 in the CT26 cancer cells. The CT26 cells with mutant IDH1 were then used to establish a cancer cachexia model and for pharmacological research. Male BALB/c mice (~7 weeks old) obtained from Shanghai SLAC Laboratory Animal Co. Ltd (Shanghai, China), were housed in an environment with a constant temperature of $22 \pm 1\,°C$ and relative humidity of $50 \pm 1\%$, with free access to food and water. After 1 week of acclimatisation, the mice were randomly divided into responsible groups, and each group contained 10 mice. A cancer cachexia group received a subcutaneous injection of 100 μl cancer cells ($1 \times 10^6$ cells) into the right flank to induce the cancer cachexia model, while an equal volume of vehicle phosphate-buffered saline (PBS) was subcutaneously injected into the right flanks of control mice. The tumors were palpable on day 9, and the mice without palpable tumors were excluded. The mice were grouped and intravenously administered with 50 mg/kg ivosidenib or PBS as NTC control every day for the following experimental period. Tumor length and width were measured every other day using a digital caliper. The tumor weight g was calculated using the formula 0.52× tumor width (cm) × length (cm)2, and the lean body weight was calculated by subtracting the tumor weight from the total animal weight. The mice were euthanized according to ethical criteria (veterinary regulatory endpoint including loss of ~25% of initial body mass). Blood was then collected in tubes, and the serum was prepared within 1 h. The tumors were dissected and weighed. The muscle gastrocnemius from both the hind legs was quickly removed and weighed. One half was snap frozen in liquid nitrogen, and the other half was fixed in 10% neutral buffered formalin.

**Orthotopic tumor model of in vivo bearing GL261 cells with IDH1-mut**. The GL261 cells with mutant IDH1 were then used to establish a cancer model and for pharmacological research. Briefly, GL261 glioma cells with mutant IDH1 were generated using lentivirus. Male C57BL/6 J mice (~7 weeks old), obtained from Shanghai SLAC Laboratory Animal Co. Ltd (Shanghai, China), were anaesthetised with a mixture of ketamine 80 mg/kg and xylazine 10 mg/kg (No.: 2021-0361). Then $3 \times 10^5$ GL261 cells with mutant IDH1 in 2 μl HBSS were implanted intracranially using a Stoelting stereotaxic apparatus (2.5 mm lateral to the bregma and 3.0 mm below the skull). Beginning on day 7 of a palpable tumor, mice were grouped and intravenously administered with 50 mg/kg ivosidenib or PBS as NTC control intravenously daily for the remainder of the experimental period. Mice were monitored daily and sacrificed at the endpoint specified by the veterinary authorities (disease or loss of ~25% of initial body mass). The muscle gastrocnemius from both the hind legs was quickly removed and weighed. One half was snap frozen in liquid nitrogen, and the other half was fixed in 10% neutral buffered formalin. The blood was then collected in tubes, and the serum was prepared within 1 h.

**Histopathology of muscle gastrocnemius**. After fixation for >24 h, the excised gastrocnemius muscle was embedded in paraffin. Transverse sections (10 μm thick) were thawed, mounted on glass slides and treated sequentially with $2 \times 5$ min' xylene, $2 \times 2$ min' 100% ethyl alcohol and $2 \times 2$ min 95% ethyl alcohol. After rinsing in distilled water, they were placed in hematoxylin (Sigma, St. Louis, MO, USA) for 1 min and rinsed first in distilled water and then in tap water for 3–5 min. Slides were rinsed in 95% ethyl alcohol for 30 s and incubated in 1% eosin for 1 min. The slides were then immersed 10 times in 95% ethyl alcohol and 10 times in 100% ethyl alcohol and cleared in 3 changes of xylene for 2 min each. Images were captured using the KF-FL-040 Auto Digital Slice Film Sweeper (Jiangfeng, Zhejiang, China). Quantification of cross-sectional area was performed using ImageJ (National Institutes of Health, Bethesda, MD). More than 100 individual myofiber areas were measured from each slide of the central field of view, which was blinded to treatments/groups.

**Metabolite extraction and quantitative determination of D2HG and 3-methylhistidine**. The analysis was performed by comparing the retention time and ionic characteristics of the HPLC–MS detection method. Standards of D2HG (H8378) and 3-methylhistidine (M9005) were purchased from Sigma-Aldrich LLC(USA). Tumor tissues (10 mg) were extracted by grinding with liquid nitrogen, and tumor fragments were homogenised in 1000 μL of pre-chilled 80% HPLC grade aqueous methanol. After centrifugation at $14,000 \times g$ for 15 min, the supernatants were transferred to polypropylene tubes and evaporated under vacuum. The medium after culturing the IDH1 mutant and wild-type cancer cells for 24 h, 48 h and 72 h was collected, centrifuged, deproteinised and concentrated. The supernatants were also transferred to polypropylene tubes and evaporated under vacuum. Residues were reconstituted with 50 μL of methanol-water (90%:10%; v/v), and 10 μL was injected into an Agilent 6490 Triple Quadrupole LC/MS system. Serum D2HG and 3-methylhistidine levels were determined after comparison and calculation of the responsible peak area.

**Statistics and reproducibility**. Data were summarized using means and standard deviation (SD). At least three independent biological replicates were performed and individual data points are shown in all graphs with the exception of myotube width and cross-sectional area. No data were excluded from the analysis. Two groups were compared using unpaired two-tailed Student's t-test. For multiple independent groups, one-way or two-way analysis of variance (ANOVA) with Tukey post hoc multiple comparison test was used. Data that did not pass the variance test were compared with nonparametric two-tailed Mann–Whitney rank sum tests, or ANOVA on ranks tests. Survival data for mice were tested for statistical significance using Kaplan–Meier curves with two-sided log-rank Mantel–Cox analysis. Relative expression was determined by comparing the treatment values to the control values after normalization to controls. Statistical significance was assessed using the GraphPad Prism 7 software. P-values <0.05 are considered as statistically significant and the significance levels are marked ****$P < 1e-4$, ***$P < 2e-4$, **$P < 2e-3$, *$P < 0.05$. NS not significant at $P > 0.05$.

**Reporting summary**. Further information on research design is available in the Nature Portfolio Reporting Summary linked to this article.

## Data availability

Source data underlying all graphs in the paper are presented in Supplementary Data 17. The uncropped and unedited blot/gel images of Figs. 3 and 5 are presented in the supplementary materials of supplementary Fig. S3 and supplementary Fig. S6. Experimental materials and sources are presented in the supplementary note. The plasmids of pLV-EF1α-FLAG-IRES-Puro (Plasmid #85132 from Addgene) were used. The RNA-seq data have been deposited in the NCBI Sequence Read Archive under accession numbers: SUB8925446 and SUB8927518. Additional data associated with this paper are available from the authors.

## Code availability

The customized code for mRNA-seq data analysis is available from the https://github.com/quanjunyang11/RNAseq_script. The standard code for metabolomics data analysis, and multi-omics analysis is available from MetaboAnalyst (https://www.metaboanalyst.ca/).

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

## Acknowledgements

The authors acknowledge the help of the Analysis and Testing Centre of Shanghai Jiaotong University and Naval Medical University (metabolomics analysis and metabolite determination) and Qiantang Biotechnology (Suzhou) Co., Ltd (transcriptomics analysis). This study was supported by the National Natural Science Foundation of China (No. 82272925, 81872494, and 81873042), Excellent Young Medical Talent Training Program of Shanghai Municipal Health Commission (2018YQ19) and the Shanghai Pujiang Talent Program (21PJ1411900).

## Author contributions

Q.Y., C.G.: designed and performed most of the experiments in the study. X.Zhu., J.Ha., R.X., D.C., T.Y., J.D., S.Zhe.: clinical samples collection and analysis. X.Zha.: performed metabolites selection, metabolomics, and transcriptome analysis. H.Z., M.C., Y.W., S.Zho.: performed cell culture, sample preparation, and molecular biology experiment. Q.Y., J.Ha., M.C., Y.W.: performed the animal experiment. X.S., J.Z.: provision of study materials, reagents, materials, and instrumentation. J.Hu., B.X.: performed data analysis and participated in technical advice. Q.Y., J.Ha., C.G.: manuscript preparation. Q.Y., C.G., X.Zhu., J.Ha.: review and editing manuscript.

## Competing interests

The authors declare no competing interests.
