## [Peer Review File · Communications Biology]

Reviewers' comments:

Reviewer #1 (Remarks to the Author):

This study by Zhu and colleagues employs complementary myotube culture and murine cancer models to explore the role of Oncometabolite D-2-hydroxyglutarate in muscle wasting and was associated with a mutation in IDH1. D-2-hydroxyglutarate was found to induce myotube atrophy that could be prevented by overexpression of D2hgdh. Using mouse models of cancer, inhibition of IDH1 could attenuate cachexia. Overall, the authors provide intriguing data supporting a mechanistic role of IDH1-derived D2hg in the development of muscle wasting, however, some methodological concerns hinder the strength of conclusions. These major concerns are described below:

Major Comments

- 1) While qPCR changes in proteasome related genes is in line with increased proteolytic activity, a major weakness of the current work is a lack of robust and direct measures of protein synthesis/degradation in the employed culture and animal models. There are numerous easily applied approaches to study muscle protein synthesis and protein degradation that would substantially strengthen the conclusions that could be drawn.
- 2) The methods describes data presentation as the mean +/- SD. Are the authors sure this is the approach used in all graphs? Several graphs appear more like the error bars represent the SEM.
- 3) In some cases, sufficient methodological detail is lacking from the manuscript. For example, the analysis of gastro muscle histopathology is not described. Was the muscle area or individual myofiber area measured? Was this done manually? Was the analyzed blinded to treatments/groups?
- 4) Based on the methods, it appears that animal randomization and blinding of treatments was not performed. This must be considered a major weakness and diminishes enthusiasm for the conclusions drawn substantially. An addition, male mice are solely employed for a strong scientific justification for the exclusion of female mice is not provided, despite female patients experiencing cachexia similar to males.
- 5) In Fig 1C performing the myotube screen, it is unclear what the individual points represent. The methods describe the use of three independent replicates, but far more than 3 points are presented. I assume these points then represent individual myotube, but not information is provided on how these were chosen. If individual myotubes, the numbers presented seem quite small for 3 biological replicates. Additionally, if individual myotubes, a nested statistical analysis should be performed as myotubes from the same sample should be considered as individual datapoints.
- 6) There is limit experimentation to determine if D2Hg/ketoglutarate are the key metabolite signals linked to muscle proteolysis or whether the proteolytic alterations or linked to redox changes in the NADPH/NADP+ or NADH/NAD+ alterations observed. This lack of mechanistic information hinders more targeted therapy development aimed to attenuate cachexia in IDH1 mutant patients.

Minor Comments

- 1) There are some grammatical errors that need correction. For example, Fig 1C units should be μM not uM (micromolar) and Figure 1D,E should be labeled at 'Expression' not 'Expresxion'
- 2) Figure 2H – units of measure are not provided. It is also unclear is this graphs is meant to show the whole muscle area or it represents the myofiber area (it would seem the latter is more likely). Related to this, if this graph is myofiber area, can the authors provide an explanation for why the magnitude of change in the IDH1-mutant vs WT cancer is larger for myofiber area compared to gastric muscle mass (panel F)?
- 3) Figure 2I – it is unusual for the relative mRNA level of TRIM63 to all be 1 in the control condition. The reviewer recognizes that the data are $2^{-\Delta\Delta\text{CT}}$, but have them all equal 1 would indicate that the

CT values were identical for each sample which would be extremely unlikely.

4) Figure 4G,I – color labeling for groups appears to be incorrect.

5) Figure 5 – volcano plots presented should be based on the adjusted p-value

6) From a presentation standpoint, the figures could use some improvement. The alignment of graphs and consistent font type/size would greatly improve the data presentation.

7) English language edited could improve the readability of the work. This is not considered a major weakness, but rather a suggestion.

Reviewer #2 (Remarks to the Author):

The authors in the manuscript titled, 'Oncometabolite D-2-hydroxyglutarate - 1 dependent metabolic reprogramming induces skeletal muscle atrophy during cancer cachexia', illustrate the IDH-associated increase in D2HG which leads to an increase in atrophy. The authors pointed towards potential mechanisms of metabolic reprogramming induced by D2HG. These are my concerns:

1. How were these manuscripts selected? Was there a systematic review of manuscripts? Were there manuscripts which were discarded?

2. Fumarate, Succinate and D2HG all contain acidic groups which are capable of decreasing the pH of the media. Was the pH measured in these treatments?

3. In figure 1, the authors first showed that fumarate was responsible for increasing the mRNA expression of Trim63 and Fbxo32. However, after that they mention that only D2HG induced myotube thinning occurs via proteolysis. Can the authors explain why do the expression of Trim63 and Fbxo32 increase with fumarate.

4. In figure 2, the authors talk about increased D2HG in the serum of cachectic vs non-cachectic patients. The authors should also mention if the 19 metabolites shortlisted in figure 1 were all increased in the muscles or the serum in the published reports.

5. The authors must collect the conditioned media from IDH1-R132H mutant cells and wildtype cells and assess the difference in the level of D2HG accumulation.

6. After implantation, the mutant cells grew tumors faster than the wild type cancer cells and also underwent body weight loss faster than the wildtype. This phenomenon might have nothing to do with the increase in D2HG but the fact that IDH is a known tumor promoter, and increases tumor growth, and increased tumor burden leads to faster loss of muscle mass. To prove causation that the effect is wholly only due increased D2HG, the tumor burden needs to be the same.

7. The authors demonstrated atrophic effect of adding D2HG to C2C12. However, if D2HG can't be catabolized by C2C12, then how is D2HG having its effect on the metabolites etc.

Reviewers' comments:

Reviewer #1 (Remarks to the Author):

This study by Zhu and colleagues employs complementary myotube culture and murine cancer models to explore the role of Oncometabolite D-2-hydroxyglutarate in muscle wasting and was associated with a mutation in IDH1. D-2-hydroxyglutarate was found to induce myotube atrophy that could be prevented by overexpression of D2hgdh. Using mouse models of cancer, inhibition of IDH1 could attenuate cachexia. Overall, the authors provide intriguing data supporting a mechanistic role of IDH1-derived D2hg in the development of muscle wasting, however, some methodological concerns hinder the strength of conclusions. These major concerns are described below:

Response: We greatly appreciate the critical reading of our manuscript entitled "Oncometabolite D-2-hydroxyglutarate-dependent metabolic reprogramming induces skeletal muscle atrophy during cancer cachexia" and thank you for your valuable suggestions and comments. We have carefully considered the comments and revised the manuscript accordingly. The responses to the comments are listed below (see below). We have indicated in blue where we have incorporated the revised manuscript.

Major Comments

1) While qPCR changes in proteasome related genes is in line with increased proteolytic activity, a major weakness of the current work is a lack of robust and direct measures of protein synthesis/degradation in the employed culture and animal models. There are numerous easily applied approaches to study muscle protein synthesis and protein degradation that would substantially strengthen the conclusions that could be drawn.

Response: Thank you for your valuable comments and suggestions. Protein degradation was determined by Western blot in Figure 2J and 2K based on the major ubiquitinated protein degradation pathway. The biomarker of protein degradation was also reflected by the release of 3-methylhistidine as shown in Figure S11B. Protein synthesis was determined by Western Blot of phosphorylation levels of mTOR, P70S6K and 4E-BP1 in Figure 2L and Figure S11C.

2) The methods describes data presentation as the mean +/- SD. Are the authors sure this is the approach used in all graphs? Several graphs appear more like the error bars represent the SEM.

Response: Thank you for your valuable comments. The statistical analysis has been revised in the manuscript.

3) In some cases, sufficient methodological detail is lacking from the manuscript. For example, the analysis of gastro muscle histopathology is not described. Was the muscle area or individual myofiber area measured? Was this done manually? Was the analyzed blinded to treatments/groups?

Response: Thank you very much for your valuable comments. The method has been revised. Specifically, the histopathology of the gastrocnemius muscle has been revised in the Methods section of the manuscript. More than 100 individual myofiber areas were measured from each slide of the central field of view. The individual myofiber areas were measured manually which was blinded to treatments/groups.

4) Based on the methods, it appears that animal randomization and blinding of treatments was not performed. This must be considered a major weakness and diminishes enthusiasm for the conclusions drawn substantially. An addition, male mice

are solely employed for a strong scientific justification for the exclusion of female mice is not provided, despite female patients experiencing cachexia similar to males.

Response: Thank you for your valuable comments. After one week of acclimatization, the mice were randomly divided into responsible groups, and each group had 10 mice. The average body weight of each group was similar with no significant difference. We have published several papers on this mouse cancer cachexia model since 2013, and confirmed the same process of cancer cachexia in male and female mice. We had discussed this limitation in the revised manuscript.

5) In Fig 1C performing the myotube screen, it is unclear what the individual points represent. The methods describe the use of three independent replicates, but far more than 3 points are presented. I assume these points then represent individual myotubes, but not information is provided on how these were chosen. If individual myotubes, the numbers presented seem quite small for 3 biological replicates. Additionally, if individual myotubes, a nested statistical analysis should be performed as myotubes from the same sample should be considered as individual datapoints.

Response: Thank you very much for your valuable comments. The individual dots in Figure 1C represent individual myotube widths. We performed three independent biological replicates in a six-well plate. We measured approximately 10 myotubes per well. The width of each individual myotube was measured at the widest cross section, and each myotube was measured in three different regions (the average was the width). Each group contained 3 biological duplicates and approximately 30 individual myotube widths were measured. The nested statistical analysis was repeated for the analysis of Figure 3C as “Nested One-way ANOVA with post hoc Tukey's multiple comparison tests”

6) There is limit experimentation to determine if D2HG/ketoglutarate are the key metabolite signals linked to muscle proteolysis or whether the proteolytic alterations or linked to redox changes in the NADPH/NADP⁺ or NADH/NAD⁺ alterations observed. This lack of mechanistic information hinders more targeted therapy development aimed to attenuate cachexia in IDH1 mutant patients.

Response: We thank you for your valuable comments. D2HG-induced skeletal muscle protein proteasomal changes were shown in Figure 2I, 2J, and 2K from the in vivo experiment and Figure S3C, S3D from the in vitro experiment. In the in vivo experiment of IDH1 mutant tumor-bearing mice, there was a high serum D2HG concentration (Figure 2M) and increased protein and gene expression of E3 ligases of Trim63/MuRF1 and Fbxo32/Atrogin-1 in gastrocnemius muscle. The higher D-2-hydroxyglutarate group also showed high ubiquitin expression. The in vitro experiment of multinucleated myotubes treated with D2HG showed increased gene expression of E3 ligases of Trim63 and Fbxo32. We measured the concentration of NADH and NAD⁺ and it was interesting that D2HG induced an increased NAD⁺/NADH redox ratio.

Minor Comments

1) There are some grammatical errors that need correction. For example, Fig 1C units should be μm not μM (micromolar) and Figure 1D,E should be labeled at ‘Expression’ not ‘Expresxion’

Response: Thank you for your suggestion. Figure 1 has been revised in the manuscript.

2) Figure 2H – units of measure are not provided. It is also unclear if this graph is meant to show the whole muscle area or it represents the myofiber area (it would seem the latter is more likely). Related to this, if this graph is myofiber area, can the authors provide an explanation for why the magnitude of change in the IDH1-mutant vs WT cancer is larger for myofiber area compared to gastric muscle mass (panel F)?

Response: Thank you for your comments and suggestions. The units of Figure 2H were μm^2 , and we have adjusted the unit.

3) Figure 2I – it is unusual for the relative mRNA level of TRIM63 to all be 1 in the control condition. The reviewer recognizes that the data are $2^{-\Delta\Delta\text{CT}}$, but having them all equal 1 would indicate that the CT values were identical for each sample which would be extremely unlikely.

Response: Thank you for your comments and suggestions. The control expression has been updated after reviewing the original data.

4) Figure 4G,I – color labeling for groups appears to be incorrect.

Response: Thank you for your comment. The color labeling has been updated.

5) Figure 5 – volcano plots presented should be based on the adjusted p-value.

Response: Thank you for your comments and suggestions. The methods and results have been revised in the manuscript. The markers have been updated and the data have been added to the Supplementary Material.

6) From a presentation standpoint, the figures could use some improvement. The alignment of graphs and consistent font type/size would greatly improve the data presentation.

Response: Thank you for your comments and suggestions. The methods and results have been revised in the manuscript.

7) English language edited could improve the readability of the work. This is not considered a major weakness, but rather a suggestion.

Response: Thank you for your comments and suggestions. The English language in the manuscript has been revised. The grammar, spelling, and punctuation errors have been corrected.

Reviewer #2 (Remarks to the Author):

The authors in the manuscript titled, ‘Oncometabolite D-2-hydroxyglutarate - 1 dependent metabolic reprogramming induces skeletal muscle atrophy during cancer cachexia’, illustrate the IDH-associated increase in D2HG which leads to an increase in atrophy. The authors pointed towards potential mechanisms of metabolic reprogramming induced by D2HG. These are my concerns:

1. How were these manuscripts selected? Was there a systematic review of manuscripts? Were there manuscripts which were discarded?

Response: We greatly appreciate the critical reading of our manuscript entitled "Oncometabolite D-2-hydroxyglutarate-dependent metabolic reprogramming induces skeletal muscle atrophy during cancer cachexia" and thank you for your valuable comments. Clinical evidence has shown that some types of solid tumors seem to be

predisposed to cachexia, and we designed the experiment for several years. The present study investigated cancer-related metabolites in the development of cachexia by screening for active metabolites and examining their effect on muscle wasting. All the established experimental data obtained revealed the contribution of IDH1 mutation in mediating excessive D2HG accumulation in cancer cachexia progression and highlight the possibility of individualized treatment for cancer cachexia patients with IDH1 mutation. The manuscript was submitted to Nature Communication for review under manuscript number NCOMMS-22-01978-T. After 2 months of review, the manuscript was rejected and we repeated the experiment and added more data for the manuscript.

2. Fumarate, Succinate and D2HG all contain acidic groups which are capable of decreasing the pH of the media. Was the pH measured in these treatments?

Response: This is a good suggestion. We repeated the experiment and prepared the fusion medium (DMEM with 2% horse serum and 1% penicillin/streptomycin) containing fumarate (50 μ M), succinate (50 μ M) and D2HG (93 μ M). The pH was then measured and the pH distribution was within the normal range of 7.08 and 7.51.

3. In figure 1, the authors first showed that fumarate was responsible for increasing the mRNA expression of Trim63 and Fbxo32. However, after that they mention that only D2HG induced myotube thinning occurs via proteolysis. Can the authors explain why do the expression of Trim63 and Fbxo32 increase with fumarate.

Response: Thank you for your suggestions. From our metabolite screening, we found that fumarate treatment resulted in increased gene expression of the E3 ligases Trim63 and Fbxo32. Trim63 and Fbxo32 are E3 ligases responsible for the ubiquitinated proteasome pathway that mediates protein turnover. Previous studies have showed that dimethyl fumarate regulates Fbxo32 expression (Meseguer-Ripolles J, Lucendo-Villarin B, Tucker C, et al. Dimethyl fumarate reduces hepatocyte senescence following paracetamol exposure[J]. *Iscience*, 2021, 24(6): 102552). These results led us to repeat the experiment and we found the increased expression of Trim63 and Fbxo32 after treatment with 50 μ M fumarate. The mechanism of muscle wasting was complex and heatmap of differentially expressed genes after D2HG and fumarate treatment of well-differentiated myotubes showed different transcriptional profile.

4. In figure 2, the authors talk about increased D2HG in the serum of cachectic vs non-cachectic patients. The authors should also mention if the 19 metabolites short listed in figure 1 were all increased in the muscles or the serum in the published reports.

Response: Thank you for your comments and suggestions. The 19 metabolites in Figure 1B are taken from Table S1 and were obtained from published articles on cancer cachexia patients. We used an in vitro experiment to screen for these metabolites that mediate muscle atrophy. The alteration of these metabolites occurred in patients with specific mutated oncogenes. D2HG was increased in cancer cachexia patients with IDH1 mutation.

5. The authors must collect the conditioned media from IDH1-R132H mutant cells and wildtype cells and assess the difference in the level of D2HG accumulation.

Response: Thank you for your comments and suggestions. We measured and compared the D2HG concentration of IDH1-R132H mutant and wild-type cells, and the results are shown in Figure S2D. There was no accumulation of D2HG in the medium at 24, 48, and 72 hours after culturing the wild-type CT26 and GL261 cells. In addition, there

was a gradual accumulation of D2HG at 24, 48, and 72 hours after culturing the IDH1 mutant CT26 and GL261 cells.

6. After implantation, the mutant cells grew tumors faster than the wild type cancer cells and also underwent body weight loss faster than the wildtype. This phenomenon might have nothing to do with the increase in D2HG but the fact that IDH is a known tumor promoter, and increases tumor growth, and increased tumor burden leads to faster loss of muscle mass. To prove causation that the effect is wholly only due increased D2HG, the tumor burden needs to be the same.

Response: Thank you very much for your valuable comments. Regarding the tumor weight curve, there were no changes between the two groups of CT26 cancer cells with/without IDH1 mutation. While the lean body weight of CT26 tumor-bearing mice with IDH1 mutation was significantly lower than that of wild-type CT26 tumor-bearing mice from DPI 22. Cachexia was defined as a loss of lean body weight (mice without transplanted tumor) of more than 5% from the lean body weight change curve. In the IDH1-R132H mutant cancer-bearing mice, cancer cachexia syndrome occurred at DPI 17, and the average lean body weight decreased by 5.4% (from 26.83±1.12 g to 25.45±0.98 g) (Figure 2E). However, cachexia was observed in the wild-type tumor group at DPI 22, as the lean body weight decreased by 5.0% (from 27.09±1.10 g to 25.81±1.23 g). These results indicated that mutation of IDH1 in CT26 cells could accelerate tumor growth and induce body weight loss.

7. The authors demonstrated atrophic effect of adding D2HG to C2C12. However, if D2HG can't be catabolized by C2C12, then how is D2HG having its effect on the metabolites etc.

Response: Thank you very much for your valuable comments. The accumulation of D2HG was found to be associated with the onset of cancer cachexia. We also found that the high dosage of 93 μ M D2HG in the medium resulted in muscle wasting. The mechanism was reflected in the upregulation of the ubiquitinated protein system and metabolic reprogramming. Differential transcriptional profiling (Figure 3B) showed that D2HG treatment resulted in 412 transcriptional changes based on fold change at 2.0 and adjusted q at 0.05 (Table S7). To reveal the primary mechanism responsible for the catabolic pathway, overrepresentation analysis (ORA) was used to screen the different genes. ORA-based gene ontology (GO) enrichment revealed altered molecular function, biological process, and cellular component based on up- or down-regulated genes (Figure 3C). Extracellular matrix structural component, structural molecule activity, cytoskeleton structural component, oxidoreductase activity, acting on the CH-OH group of donors NAD or NADP as acceptor, glutathione transferase activity, and NAD binding were the top molecular functions. In addition, the biological process and cellular components, including muscle cell differentiation, muscle tissue development, muscle system process, and muscle contraction, were also disturbed after D2HG treatment (Figure S4). These results indicated that D2HG may act as complex metabolic mediator that disturbs C2C12 metabolism.

Reviewers' comments:

Reviewer #1 (Remarks to the Author):

Overall the authors have thoughtfully responded to the major criticisms raised on my previous review. The addition blots related to markers of protein degradation are a welcomed addition, although I would strongly encourage the authors to discuss the limitations as these new data are still not direct or rigorous measures of protein synthesis or degradation. Moreover, only representative blots are presented and quantified data are lacking so conclusions cannot be drawn from these newly added data. Below I still have several concerns that should be addressed:

1) The statistical analysis section is still ambiguous and would seemingly indicate a lack of careful attention to details. Several issues include the following:

-Presenting data as either SD or SEM is inappropriate. The authors should choose one approach (SD being more relevant in this reviewer's opinion) and make this consistent for all data (aside from omic).

-paired t-test should only be used when samples are being compared from the same source (i.e., animal), not between two groups.

-The nested ANOVA details are not described aside from simply saying this in the figure legend.

-For t-test and ANOVA, normally distributed data are required but there is not described testing for data normality.

2) Still the lack of discussion of using only male mice remains absent. The authors have added the following sentence: "Male mice showed a similar progression to female mice, so we used male mice for the following in vivo experiment", but provide no citation or data to support this statement. The assumption that males and females are the same is unequivocally difficult to support scientifically. While I recognize that animal protocol limitations at their institution may be driving this decision, this should be thoroughly discussed as a limitation and appropriate justification should be provided in the manuscript. Otherwise, this paper simply adds to the existing literature that suppresses biological sex as an important experimental variable for consideration.

Reviewer #2 (Remarks to the Author):

The authors responded and revised the manuscript to the best of their abilities. My concern #1 was still not answered, rather it was a very general and vague answer. I asked how were the manuscripts selected in Table S1 from which 157 cachexia metabolites were shortlisted. Otherwise, the manuscript is stronger now.

Reviewers' comments:

Reviewer #1 (Remarks to the Author):

Overall the authors have thoughtfully responded to the major criticisms raised on my previous review. The addition blots related to markers of protein degradation are a welcomed addition, although I would strongly encourage the authors to discuss the limitations as these new data are still not direct or rigorous measures of protein synthesis or degradation. Moreover, only representative blots are presented and quantified data are lacking so conclusions cannot be drawn from these newly added data. Below I still have several concerns that should be addressed:

Thank you very much for your valuable comments and suggestions. Protein synthesis and degradation have been discussed in the revised manuscript. The quantified data of protein expression have been included (Fig. 3 and Fig. S11). "The increasing gene and protein expression of the E3 ligases *Trim63* (MuRF1) and *Fbxo32* (*Atrogin-1*) reflects increased protein degradation. Whereas the decreasing protein expression of phosphorylated levels of mTOR, P70S6K and 4E-BP1 indicates attenuated protein synthesis." Results and Discussion have been revised.

The main text should not exceed 5000 words. We have revised the manuscript according to your suggestions. We have carefully reviewed the comments and revised the manuscript accordingly. The responses to the comments are listed below.

Figure 3

Figure 3. IDH1 mutation-mediated D2HG accumulation induced a decrease in muscle protein synthesis and an increase in muscle protein proteolysis. (A) Column plot of Fbxo32 and Trim63 mRNA expression in IDH1 mutant cancer-bearing mice and wild-type cancer-bearing mice. One-way ANOVA with post hoc Tukey's multiple comparison test (**** $p < 1e-4$, *** $p < 2e-4$, ** $p < 2e-3$, * $p < 0.05$). (B-C) The protein expression and quantification data of the degradation-related E3 ligases atrogin-1 and MuRF1 in IDH1-mutant cancer-bearing mice and wild-type cancer-bearing mice. One-way ANOVA with post hoc Tukey's multiple comparison test (**** $p < 1e-4$, *** $p < 2e-4$, ** $p < 2e-3$, * $p < 0.05$). (D) The protein expression of degradation-related ubiquitin in IDH1 mutant cancer-bearing mice and wild-type cancer-bearing mice. (E-F) The protein expression and quantification data of synthesis-related mTOR, P70S6K and 4E-BP1 expression and phosphorylation levels in IDH1 mutant cancer-bearing mice and wild-type cancer-bearing mice. One-way ANOVA with post hoc Tukey's multiple comparison test (**** $p < 1e-4$, *** $p < 2e-4$, ** $p < 2e-3$, * $p < 0.05$). (G) Column plot of tumor and serum D2HG levels in the IDH1 mutant cancer bearing group and wild type cancer bearing mice. One-way ANOVA with post hoc Tukey's multiple comparison test (**** $p < 1e-4$, *** $p < 2e-4$, ** $p < 2e-3$, * $p < 0.05$).

Figure S11

Figure S11. The protein synthesis/degradation of gastrocnemius muscle in IDH1(R132H) mutant GL261 cancer and wild-type GL261 cancer treated with ivosidenib or NTC. (A) The protein expression and quantified data of E3 ligase atrogin-1 and MuRF1. One-way ANOVA with post hoc Tukey's multiple comparison test (****p<1e-4, ***p<2e-4, **p<2e-3, *p<0.05). (B) Serum 3-methylhistidine levels. One-way ANOVA with post hoc Tukey's multiple comparison test (****p<1e-4, ***p<2e-4, **p<2e-3, *p<0.05). (C) The phosphorylation levels and quantified data of mTOR, p70s6k and 4E-BP1. One-way ANOVA with post hoc Tukey's multiple comparison test (****p<1e-4, ***p<2e-4, **p<2e-3, *p<0.05).

1) The statistical analysis section is still ambiguous and would seemingly indicate a lack of careful attention to details. Several issues include the following:

-Presenting data as either SD or SEM is inappropriate. The authors should choose one approach (SD being more relevant in this reviewer's opinion) and make this consistent for all data (aside from omic).

-paired t-test should only be used when samples are being compared from the same source (i.e., animal), not between two groups.

-The nested ANOVA details are not described aside from simply saying this in the figure legend.

-For t-test and ANOVA, normally distributed data are required but there is not described testing for data normality.

Response: Thanks for your valuable comments. The statistics and reproducibility have been revised as follows. All source data underlying the graphs presented in the main figures were made available as supplementary data.

Data were summarized using mean and standard deviation (SD). At least three independent biological replicates were performed and single data points are shown in all graphs except for myotube width and cross-sectional area. No data were excluded from analysis. Two groups were compared using unpaired two-tailed Student's t-test. For multiple independent groups, one- or two-way analysis of variance (ANOVA) with Tukey post hoc multiple comparison test was used. Data that did not pass the variance test were compared using nonparametric two-tailed Mann-Whitney rank sum tests (for unpaired comparisons) or ANOVA on ranks tests. Survival data for mice were tested for statistical significance using Kaplan-Meier curves with two-sided log-rank Mantel-Cox analysis. Relative expression was determined by comparing treatment values to control values after normalization to controls. Statistical significance was evaluated using GraphPad Prism 7 software. P values less than 0.05 are considered statistically significant and significance levels are indicated **** $P < 1e-4$, *** $P < 2e-4$, ** $P < 2e-3$, * $P < 0.05$. NS, not significant at $P > 0.05$.

2) Still the lack of discussion of using only male mice remains absent. The authors have added the following sentence: "Male mice showed a similar progression to female mice, so we used male mice for the following in vivo experiment", but provide no citation or data to support this statement. The assumption that males and females are the same is unequivocally difficult to support scientifically. While I recognize that animal protocol

limitations at their institution may be driving this decision, this should be thoroughly discussed as a limitation and appropriate justification should be provided in the manuscript. Otherwise, this paper simply adds to the existing literature that suppresses biological sex as an important experimental variable for consideration.

Response: Thank you for your valuable comments. We have revised the manuscript regarding the use of male mice. The references have been included.

In the Methods section, we revised the sentences "Male patients and mice showed a higher prevalence of cachexia, greater weight loss, and muscle wasting compared with female cancer patients 38,47, so we used male mice for the following in vivo experiment.

In the Discussion section, we revised the limitation. "Fourth, a growing number of studies in humans and rodent models have shown sex differences in normal skeletal muscle and cancer cachexia³⁸. Male cancer patients generally have a higher prevalence of cachexia, greater weight loss or muscle wasting, and worse outcomes than female cancer patients³⁹. Therefore, we used male mice bearing CT26 colon cancer cells and GL261 glioma cells for the cancer cachexia model, as most studies have done⁴⁰⁻⁴³. Further studies to evaluate the sex differences in IDH1 mutation and cancer cachexia progression would improve the basic mechanisms and treatment strategy⁴⁴. "

Reviewer #2 (Remarks to the Author):

The authors responded and revised the manuscript to the best of their abilities. My concern #1 was still not answered, rather it was a very general and vague answer. I asked how were the manuscripts selected in Table S1 from which 157 cachexia metabolites were shortlisted. Otherwise, the manuscript is stronger now.

Response: Thank you for your valuable comments. We have revised the manuscript regarding the selection of metabolites and the content has been included in Figure 1A. The revised manuscript has been updated.

After consulting published articles ⁸⁻¹¹, we listed cancer cachexia-related metabolites and identified 157 cachexia-related metabolites (Table S1). The metabolites were aligned with the Human Metabolome Database (www.hmdb.ca) and the functional

annotations were included. To exclude metabolites that changed in only one project, we found and selected 66 common metabolites that showed consistent changes in multiple research projects (Fig. 1A). Based on the muscle-related functional annotation and the accessibility of these metabolites, nineteen candidate metabolites were selected for the *in vitro* experiment (Table S2). The concentrations of these metabolites were listed based on the reference listed in Table S2.

REVIEWERS' COMMENTS:

Reviewer #1 (Remarks to the Author):

The authors have done a nice job addressing my previous comments and the manuscript is improved in its quality and clarity. I have no further comments of a substantive nature.

Reviewers' comments:

Reviewer #1 (Remarks to the Author):

The authors have done a nice job addressing my previous comments and the manuscript is improved in its quality and clarity. I have no further comments of a substantive nature.

Response: Thank you for your comments. We will do our best to advance knowledge and serve patients. Thank you again for reviewing our manuscript.